# FINAM - is not a model (v1.0): a new Python-based model coupling framework

Sebastian Müller[1,*], Martin Lange[1,*], Thomas Fischer[1], Sara König[1], Matthias Kelbling[1], Jeisson Javier Leal Rojas[1], and Stephan Thober[1]

[1]Helmholtz Centre for Environmental Research – UFZ, Leipzig, Germany
[*]These authors contributed equally to this work.

**Correspondence:** Sebastian Müller (sebastian.mueller@ufz.de)

**Abstract.**

In this study, we present a new coupling framework named FINAM (short for "FINAM Is Not A Model"). FINAM is designed to facilitate the coupling of models that were developed as stand-alone tools in the first place, and to enable seamless model extensions by wrapping existing models into components with well-specified interfaces. Although established coupling solutions such as ESMF, OASIS or YAC focus on highly parallel workflows, complex data processing, and regridding, FINAM prioritizes usability and flexibility, allowing users to focus on scientific exploration of coupling scenarios rather than technical complexities. FINAM emphasizes ease of use for end users to create, run, and modify model couplings, as well as for model developers to create and maintain components for their models. The framework is particularly suited for applications where rapid prototyping and flexible model extensions are desired. It is primarily targeting environmental models, including ecological models for animal populations, individual-based forest models, field-scale crop models, economical models, and hydrological models. Python's robust interoperability features further enhance FINAM's capabilities, allowing to wrap and use models written in various programming languages like Fortran, C, C++, Rust, and others. This paper describes the main principles and modules of FINAM and presents example workflows to demonstrate its features. These examples range from simple toy models to well-established models like OpenGeoSys and Bodium covering features like bidirectional dependencies, complex model coupling, and spatio-temporal regridding.

## 1 Introduction

Environmental models represent specific systems or parts of the environment, such as the water cycle, the carbon cycle, or the species distribution. They are usually developed to investigate specific research questions and phenomena such as hydrological droughts or reduced plant productivity. However, some phenomena are the result of processes that are interlinked and often occur at the same time. To study these complex phenomena, it is necessary to combine independently developed models.

Coupling models means that data is exchanged between them, which can be established at several levels (Brandmeyer and Karimi, 2000). There are three main approaches to exchange data between independent models: (i) using files, (ii) using

external coupling libraries, and (iii) using modeling frameworks. A fourth option, which we will not elaborate further, is custom solutions like merging code bases of different models or rewriting these from scratch.

File-based coupling means that output files from one model are used as input for another model, and each model is run separately for the entire simulation period. The advantage of this approach is that the two models can be executed asynchronously and that there are no modifications of the models required. However, this gets impractical if the data to exchange is large, for example, if the models work with a high spatial resolution and intermediate results and states have to be saved unnecessarily for the entire simulation period. The even greater disadvantage of a file-based coupling is that dynamic feedbacks between

models require overly complex workflows to correctly manage the huge amount of data I/O. It is an infeasible approach for complex systems with a large volume of data that needs to be exchanged. To overcome these issues, other approaches have been developed.

Coupling libraries enable data exchange between independently developed models. This is achieved by adding data exchange calls to the code base of each model. The coupling library then handles the data conversion and regridding. An example of a

widely used coupling library is OASIS (short for "Ocean Atmosphere Sea Ice Soil"), particularly its latest version OASIS3-MCT together with the Model Coupling Toolkit (MCT) (Craig et al., 2017). It is a powerful library designed for climate modeling, known for its efficient parallel communication and its ability to handle high-resolution grids. Also Yet Another Coupler (YAC), a general-purpose coupling library, excels in efficient parallel communication and time synchronization, written in C with bindings to Fortran and Python supporting diverse applications (Hanke et al., 2016; Hohenegger et al., 2023). A

disadvantage of these libraries is that the coupling needs to be configured explicitly. In other words, the coupler needs to be configured correctly and, for example, does not derive data conversion from metadata of the exchanged variables. This can be an error-prone approach, especially for inexperienced users who want to focus on the scientific problem rather than the coupling implementation details. Depending on the software design, the maintenance of the data exchange calls in each model also may create additional work for model developers because they are not used in the "offline" model version.

The last approach to be mentioned is the integration of different model concepts within one larger model framework, resulting in large and complex model systems, such as earth system models that represent atmospheric, terrestrial, and marine compartments. The idea is to encapsulate processes of models in components provided by the coupling framework to have a unified data exchange mechanism. Model coupling frameworks provide a platform for researchers and practitioners to combine different models with different scales, time horizons, and disciplinary perspectives to capture complex interactions and feed-

back mechanisms between different components of a system. A well-known example of such a framework is the Earth System Modeling Framework (ESMF), which is widely used for its high-performance capabilities and standardized data structures, making it suitable for large-scale climate and weather simulations (Collins et al., 2005; Molod et al., 2015). The disadvantage of this approach is that frameworks such as ESMF, while successfully used to couple independent codes without a complete rewrite in some large-scale applications (e.g., a coupling of the atmosphere model ICON and the coastal ocean model GETM

(Bauer et al., 2021)), are generally designed to build model systems from the ground up. As a result, they may be less suitable for independently developed models with existing code bases, where significant restructuring could be required.

There are several other domain-specific coupling solutions like preCICE (Chourdakis et al., 2022), which is an open source coupling library specialized in partitioned multiphysics simulations such as fluid-structure interaction and heat transfer, or OpenPALM (Buis et al., 2006) which is specialized on complex systems and highly parallelized computations. For completeness, we also mention the Basic Model Interface (BMI) (Hutton et al., 2020), which is not itself a coupler but rather a standardized, language-agnostic interface specification that models can implement to simplify interoperability and coupling.

Although these existing coupling solutions are powerful and well-established, they primarily focus on high-performance computing environments, emphasizing parallel data processing, efficient regridding, and scalability for large-scale simulations like global climate models. This focus can pose challenges for scientists and modelers who wish to experiment with model couplings in a more flexible and accessible manner, without the mental overhead of complex setups and specialized knowledge in parallel computing.

FINAM (short for "FINAM Is Not A Model") aims to fill this gap by prioritizing usability and ease of coupling over extreme computational performance. Our goal is to enable scientists to couple models with minimal effort, allowing them to comfortably experiment with model setups and focus on scientific exploration rather than technical complexities. FINAM allows for the coupling of independently developed codes and seamless model extensions by wrapping existing models into components with well-specified interfaces.

The user can build models with components from scratch within the FINAM ecosystem, but they can also couple existing wrapped models, as detailed below. A central goal of FINAM is to enable self-descriptive coupling scripts by leveraging the power of Python while offloading computationally intensive parts to native models. FINAM allows for the bidirectional coupling of spatial models in an easy and flexible manner, enabling the exchange of data in memory. It provides a consistent interface that supports flexible coupling based on the common assumption that every model operates with a time loop at its core. This allows for straightforward model extensions written in Python, enabling rapid prototyping without the need to alter the original model source code. Python's reputation as a "glue" language is well established, a characteristic that stems from its robust interoperability features. This compatibility is based on a suite of libraries, which facilitate the development of wrappers to integrate models regardless of their native programming languages, as detailed in Section 2.2.

Within the following, we first describe the main principles and modules of FINAM, and then give examples for workflows to show some of the features of FINAM including bidirectional coupling, complex model coupling, and spatio-temporal regridding. We further discuss future extensions and possible applications.

## 2 Design

### 2.1 Principles

The core idea for FINAM is that existing models are wrapped in components with a well-specified interface to get and set data and to update the respective model by one internal time step. Multiple *Components* that are linked to each other, potentially using *Adapters*, are called a *Composition* in FINAM. A composition can be executed, and automatically manages updates of the coupled models and the exchange of data between them.

This concept generally makes it straightforward for developers to wrap well-structured models (see Section 3.2 for details) and, once wrapped into a FINAM component, for users to set up and run compositions. Consequently, components can be developed in isolation without detailed knowledge of the potential coupling partner models. Models can have their own temporal and spatial resolution, whereby FINAM mediates between them without user interaction.

There are multiple ways to couple models, like merging their code bases, rewriting them using a specific framework like
ESMF, modifying source code with getters and setters from a native coupler like OASIS or YAC, or by exchanging files.

Compared to these approaches, FINAM does not require framework-specific code in the models except for some very basic, generally useful functionality (see Section 2.2). In addition, end-users need only minimal knowledge about coupled models, while the specifics are all managed automatically. Finally, using Python as the common glue language allows for coupling of models in virtually any programming language. These features may not be unique individually, but their combination makes
FINAM a flexible and easy-to-use solution for coupling environmental and other models.

As a guiding light, we will use a simple but fully functional FINAM composition shown in Fig. 1. In this example, a model is coupled with a NetCDF reader to get the input data necessary to estimate the potential evapotranspiration (PET) using the Hargreaves-Samani method (H. Hargreaves and A. Samani, 1985), and with a NetCDF writer to store the results. The actual source code for this example can be found at https://git.ufz.de/FINAM/finam-examples under the *01_Hargreaves-Samani*
folder.

### 2.1.1 Components and adapters

Components are the main building blocks of a FINAM *Composition*. Each component encapsulates a self-contained piece of logic. Typically, a component represents a simulation model that is prepared for FINAM by providing the required interface like `pet` in Fig. 1. But other types of components are also possible, such as I/O components to read and write files (like
`reader` and `writer` in Fig. 1), real-time visualizations, or statistical models.

As indicated in Fig. 2, components can have an arbitrary number of inputs and outputs, handled by *input slots* and *output slots*, respectively. In addition, components can have an optional internal time step. Examples of components with a time step are simulation models and components for reading time series data. Examples of components without a time step are statistical models, static data providers, or analytical models. Components without a time step can be push-based or pull-based. This
means that they are executed when new input becomes available, or when an output is requested, respectively.

The components are linked through their input and output slots, potentially involving adapters (Fig. 3). Adapters are similar to components in that they encapsulate a piece of logic in a self-contained way. In contrast to components, an adapter always has exactly one *input slot* and one *output slot* and does not have a time step (see Fig. 2). Adapters transform the data that is passed between components. Examples for adapters are the regridding from one grid specification to another, or time interpolation
and integration (see also section 3.3). Multiple adapters can be chained if needed.

The components and adapters use a unified interface required by FINAM. Both are created by implementing these respective interfaces, where FINAM provides abstract base classes for adapters and components with and without a time step. These base classes implement the interface partially, so developers can focus on the decisive code while leaving the boilerplate to the base

```
import datetime as dt
import finam as fm
import finam_netcdf as fm_nc
from component import PET
# config
start_time = dt.datetime(1990, 1, 1)
end_time = dt.datetime(1991, 1, 1)
8    day = dt.timedelta(days=1)
# components
pet = PET(start_time=start_time, step=day)
reader = fm_nc.NetCdfReader("data/temp.nc")
writer = fm_nc.NetCdfTimedWriter(
"results/pet.nc", inputs=["PET"], step=day)
# composition
composition = fm.Composition([pet, reader, writer])
# connections
reader.outputs["tmin"] >> pet.inputs["Tmin"]
reader.outputs["tmax"] >> pet.inputs["Tmax"]
reader.outputs["lat"] >> pet.inputs["lat"]
pet.outputs["PET"] >> writer.inputs["PET"]
# execution
composition.run(end_time=end_time)
```

**Figure 1.** A simple but fully working FINAM model consisting of three components: a `reader` providing data from a NetCDF file, a `writer` storing results in a NetCDF file, and a `pet` (potential evapotranspiration) calculator. The model calculates PET from the minimum and maximum daily air temperature and latitude values using the Hargreaves–Samani method (H. Hargreaves and A. Samani, 1985). First, the components are created, then the composition is defined, next the data connections are established, and finally the composition is executed.

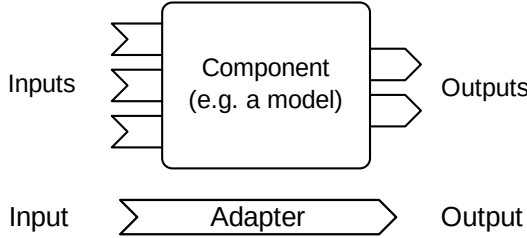

**Figure 2.** Schema of a component and an adapter in FINAM.

classes. A minimal implementation requirement for a component is illustrated in Fig. 4. The required component methods
reflect FINAM's core idea of a wrapped model. There needs to be a routine to initialize the model (`_initialize`), connect it to other models it should exchange data with (`_connect`), and a routine to update (`_update`) the model for one internal time step. In order for FINAM to properly schedule the execution of the composition, a component must provide information about its estimated next time step (`_next_time`).

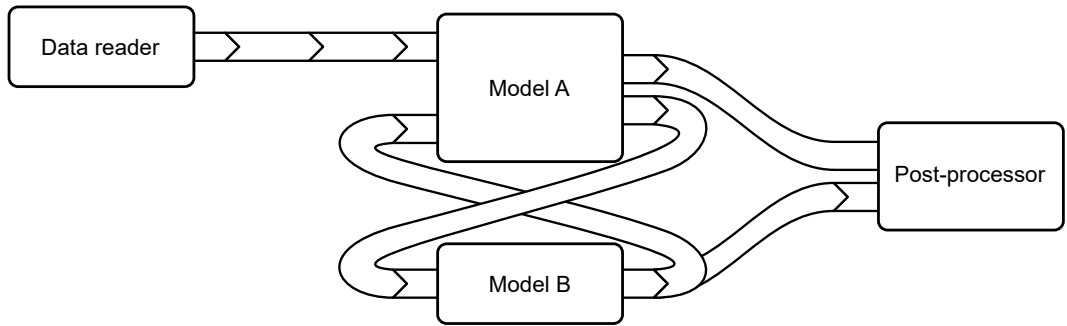

**Figure 3.** Example of a complex composition schema build with four components and several adapters.

```
class PET(finam.TimeComponent):
def _next_time(self): ...
def _initialize(self): ...
def _connect(self): ...
def _update(self): ...
```

**Figure 4.** Pseudo code for implementing the PET component from the example above in FINAM. The `_initialize` method configures the model, `_connect` prepares I/O slots and sets initial data, and `_update` executes a single time step of the model. The `_next_time` method returns the predicted simulation time of the next data pull.

### 2.1.2 Linking components

Components are linked through their input and output slots, with or without one or more adapters in between. For visual reasons, we overrode the bit shift operator ">>" to create links between an output of one component and an input of another component. This makes the coupling configuration more readable compared to chained calls of linking methods. The code example in Fig. 5 shows how two models are linked via a regridding adapter. Note that adapters, such as regridding or time interpolation, can be chained in place for minimal coding effort and readability. In Fig. 1 the `reader`, `writer` and `pet`

component are linked in lines 18-21.

```
1   hydro["runoff"] >> RegridLinear() >> stream["runoff"]
```

**Figure 5.** Data connections are denoted by the overridden bit shift operator ">>" (for visual reasons).

Data exchange between linked components and adapters takes place purely in memory, and no files are used here. During the initialization process, the compatibility of the coupled slots is checked (see Section 2.1.3), and an error is raised in case of incompatibility.

### 2.1.3 Data and metadata

For all data exchanged, FINAM uses NumPy arrays(Harris et al., 2020), wrapped in quantities provided by the `pint`[1] library, which handles units automatically. This means that any exchanged data always has units, which can, however, be `dimensionless`.

```python
def _initialize(self):
    self.inputs.add(
        name="tmin", time=self.time,
        grid=None, units="degC")
    ...
```

**Figure 6.** Excerpt from the `_initialize` method of the PET component from the example above. An input `"tmin"` for minimum temperature is created with metadata like units. The associated grid specification is undetermined at this point and will be inferred from the metadata of the connected output in the connect phase.

Each coupling slot has associated metadata and a time stamp given by a `datetime` instance if it is not static data. Since we use the built-in `datetime` module of Python, we require all models to provide their temporal data on a Gregorian calendar which is a reasonable restriction for environmental models. Obligatory metadata are grid specification and units. Grid specification types provided by FINAM allow for spatial and non-spatial data. Spatial data can be defined on structured and unstructured grids (i.e. meshes) in up to three dimensions. In addition to a grid specification and units, each coupling slot can have arbitrary custom metadata fields. All metadata follows the NetCDF Climate and Forecast (CF) metadata conventions (Hassell et al., 2017).

During the connection phase, the compatibility of linked slots is checked with respect to their grid specifications and units. Data with compatible units such as K and $^\circ$C will be automatically converted. Equivalent units such as $L/m^2$ and $mm$ will not cause a conversion. If slots are not compatible regarding grids or units, an error is raised. However, adapters can be used for transformations between different grids.

### 2.1.4 Scheduling algorithm

Component updates are scheduled by a central algorithm that decides which components will be updated next. A model update, triggered by the scheduling algorithm, involves at least three steps: (i) advance the internal time step to the next one, (ii) retrieve the input data for the current time step, and (iii) calculate the outputs for the current time step and push them to notify downstream components. An example of a component update method is given in Fig. 7.

In particular, the scheduling algorithm ensures that the required data for a component's next time step is available. For that sake, all time components must be able to report their current simulation time, as well as the (latest) expected time after the

---

[1]https://pint.readthedocs.io

next update, as this is the latest target time for the inputs of the component. An example of an estimated next time stamp is given in Fig. 8.

```python
def _update(self):
    # Increment model time
    self.time = self.next_time
    # Retrieve inputs
    data = {inp: self[inp].pull_data(self.time)
            for inp in self.inputs}
    # calculate PET for the current time step
    pet = self._calc_pet(data)
    # Push model state to outputs
    self.outputs["PET"].push_data(pet, self.time)
```

**Figure 7.** The `_update` method of the PET component from the example above. First, the component time is updated and the current required input is pulled. Then, PET is calculated and pushed to the output.

In each iteration, the scheduler starts with the component with the earliest current time and recursively analyzes its dependencies. The upstream components (the "provider") are then updated before the downstream components (the "consumer"). This ensures that the data required for the forthcoming time step is available, instead of, e.g. outdated data from the time step before.

```python
def _next_time(self):
    return self.time + self.step
```

**Figure 8.** The `_next_time` method of the PET component from the example above. Since this component uses a fix time step, we simply advance the current model time by this step.

Fig. 9 illustrates the scheduling algorithm with three components in FINAM. In panel a), A is selected as the active component because it is most back in time. The next pull time is determined, denoted by the hollow dot. A depends on B, which is not yet at A's next time, and thus becomes the active component. In panel b), B is the active component. Its next pull time is determined, again denoted by the hollow dot. B depends on C, which is not yet at B's next time, and thus becomes the active component. In panel c), C is the active component. It has no dependencies and can thus be updated. Steps a) to c) are repeated until C catches up with B's next time. In panel d), all dependencies of B (i.e. C) have sufficiently advanced in time for B to update. As illustrated by the curly braces, it is guaranteed that the input data for B is available. Any kind of interpolation between adjacent source component time steps can be applied to derive the input date. This is a responsibility of the adapters. Particularly for components with large time steps, it is also possible to integrate over multiple source component time steps. For example, component B could use the weighted average of the several steps C has performed since the last updates of B.

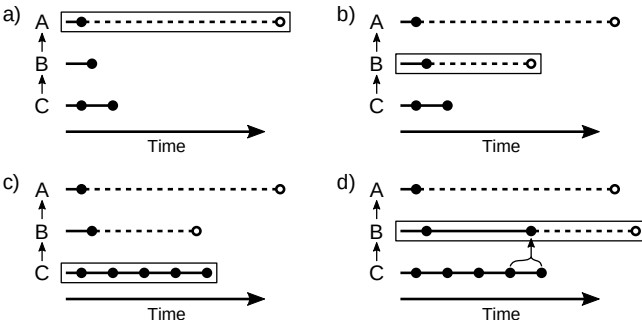

**Figure 9.** Illustration of the FINAM scheduling. Snapshots of a simulation featuring three components A, B and C with different time steps are shown. Component A depends on B, and B depends on C. Solid lines and dots denote already simulated model steps. The right-most solid dot of each component shows it's current simulation time. Dashed lines and hollow dots show the predicted next pull time of a component. The box denotes the active component.

The scheduling algorithm is mainly required for two reasons: First, it allows for the coupling of models with arbitrary, potentially incompatible time steps, and even for model time steps varying over the course of a simulation run. Second, it allows models to use input for the upcoming time step instead of the past one.

The update scheme explained so far only works if there are no cycles in the dependencies. In the case of circular or bidirectional coupling, one of the involved components must use data from the past or extrapolate in time. FINAM provides dedicated adapters that resolve circular dependencies through delayed data usage or time extrapolation. This gives users full control over how circular dependencies are resolved.

As mentioned earlier, there are also components without an internal time step. These can be updated either on pull by another
component, or on push. This allows for components like push-based file output or visualizations, or pull-based parametric data generators.

### 2.1.5   Iterative initialization

The initialization of the components may depend on other components. Possible examples are: (i) components depending on grid specification from a data source (an I/O component), (ii) deduction of the regridding transformation from input and output
grid specifications, or (iii) transfer of units of measurements from or to components that perform generic operations.

All of these examples require the exchange of metadata between components (and adapters), potentially in both directions. To make this possible in an automated way and without requiring a user to manually set all metadata, FINAM uses an iterative initialization process (Fig. 10). In this way, metadata can be exchanged downstream and upstream, regardless of complex dependencies, as long as dependency cycles can be resolved.

Implementation-wise, this metadata exchange is realized by calling the `connect` method of components multiple times (see Fig. 11 for an example). Each time the method is called, the component can try to send or obtain metadata to/from its slots.

```
def _initialize(self):
...
self.outputs.add(name="PET")
self.create_connector(
pull_data=self.inputs.names,
out_info_rules={"PET": [
FromInput("tmin", ["grid", "time"]),
FromValue("units", "mm/day")]})
```

**Figure 10.** Another excerpt from the `_initialize` method of the PET component from the example above. An output `"PET"` is created along with a connector that is configured with rules to determine metadata for this output from given inputs (grid and time specification from `"tmin"`) or hard coded values (units set to `"mm/day"`).

```
def _connect(self, start_time):
push_data = {}
if (self.connector.all_data_pulled
and self.connector.data_required["PET"]):
push_data["PET"] = self._calc_pet(
self.connector.in_data)
self.try_connect(
start_time=start_time,
push_data=push_data)
```

**Figure 11.** The `_connect` method of the PET component from the example above. The initial value for `"PET"` is calculated as soon as all required inputs are available.

Components indicate their connect progress (ready, something exchanged, or nothing exchanged) to the scheduler, which can thereby detect unresolvable dependency cycles. In this process, initial data exchange before the first time step is also handled. The procedure is largely automated through a helper class `ConnectHelper` to minimize the effort required by component developers. An example of the setup of such a connector is given in Fig. 10. FINAM users who just write coupling scripts do not need to deal with the connect phase at all.

## 2.2 Wrapping models

Wrapping an existing model requires (i) providing Python bindings for it and (ii) implementing FINAM's `Component` interface. Python bindings for an existing model need at least these features: (i) instantiate/initialize the model (ii) update the model by one time step, (iii) access state variables desired as outputs, and (iv) alter state variables desired as inputs. If no Python bindings of the model exist, but it can be run as a black-box for a single time-step, there is also the possibility to create a component that prepares the required input files for each time-step, calls the model, and reads the output files to provide the data in the FINAM composition. But be aware that this approach may introduce performance bottlenecks since it is basically a file based coupling.

Depending on the code structure of the model, some refactoring may be required to provide separate routines for initialization and model stepping that are then exposed to the Python side with Python bindings. Python bindings can be created using libraries such as Cython (Behnel et al., 2011), scikit-build (Fillion-Robin et al., 2018), f2py (Harris et al., 2020), pybind11 (Jakob et al., 2017), pyo3[2], ctypes[3], swig[4], or cffi[5].

    As an example, we want to discuss the Python bindings for the mesoscale hydrological model - mHM (Samaniego et al.,
2010; Kumar et al., 2013) written in Fortran. The developers had to take these steps: (i) encapsulating the time loop in a callable subroutine by copying over code, (ii) encapsulating the initialization and finalization of the model by separating the main driver of mHM into callable subroutines, and (iii) writing an f2py wrapper that links against the mHM library and provides routines to call the mentioned initialization, update, and finalization subroutines as well as routines to retrieve and alter the internal states. This straightforward refactoring was done in a manageable amount of commits and a positive side effect is that mHM is now
installable via pip[6].

    Using Python bindings of a model, the actual wrapper implementing the `component` interface can be written with as few as 50 lines of code for a simple use case similar to the implementations of the PET example above, where the interaction with the wrapped model would be analogous to line 8 in Fig. 7. The mHM wrapper for comparison is provided by a separate Python package `finam-mhm`[7]. At its core, the mHM component is implemented in the same way as in the PET example above but
with more inputs and outputs. FINAM's extensive documentation[8] provides a detailed guide and examples for this task.

## 2.3 Modules

Following Python's "batteries included" philosophy, the FINAM core package, along with its external packages, provides a wide array of components and adapters designed to simplify common tasks in environmental modeling. This modular approach ensures that users can install only the components they need, avoiding unnecessary dependencies and keeping their environment
clean and efficient.

    FINAM includes regridding adapters based on robust libraries such as SciPy (core module) and ESMF (proved as separate package `finam-regrid`), facilitating spatial data transformation between different grids. Temporal interpolation and integration adapters align data from models operating at different time resolutions, ensuring coherent temporal data integration.

    The framework supports file input/output (I/O) for commonly used formats in environmental modeling, such as NetCDF,
VTK, and CSV, simplifying data exchange and storage. Live plotting capabilities provided by `finam-plot` and `finam-graph`, which are based on `matplotlib`, enable real-time visualization of time series and spatial data, which is useful for monitoring ongoing simulations and making immediate adjustments. Additionally, live visualizations for scheduling and coupling composition provide an intuitive understanding of model interactions.

---

[2]https://github.com/PyO3/pyo3

[3]https://docs.python.org/3/library/ctypes

[4]https://www.swig.org

[5]https://github.com/python-cffi/cffi

[6]https://git.ufz.de/mhm/mhm/-/tree/v5.13.1/pybind

[7]https://git.ufz.de/FINAM/finam-mhm

[8]https://finam.pages.ufz.de/finam/finam-book/development/components

To address dependency cycles, where components rely on each other's data, FINAM includes adapters designed to break these cycles, ensuring that one component uses past data to maintain simulation integrity. For rapid prototyping and testing, FINAM offers components like noise generators and generic transformations, allowing simulation of various scenarios and validation of model behavior without altering the core source code.

Further, these components are provided with the core package:

- `CsvReader` and `CsvWriter` for working with CSV files

- `SimplexNoise` for generating synthetic (test) data

- `ParametricGrid` for generating synthetic (test) data from a user function

- `CallbackGenerators` and `CallbackComponent` for generating and transforming data using user-defined functions

Additional adapters in the core package include:

- `RegridLinear` and `RegridNearest` for regridding based on SciPy

- `LinearTime`, `StepTime`, `AvgOverTime` and `SumOverTime` for time interpolation and aggregation

- `GridToValue`, `ValueToGrid` and `Histogram` for conversion between gridded and non-gridded data

- `Callback` for arbitrary conversions based on a user function

By integrating these components, FINAM enhances its utility and flexibility, making it a powerful tool for environmental modelers. The modular design simplifies installation and configuration, supporting the development and execution of complex simulations with greater ease and efficiency. This comprehensive suite of tools underscores the commitment of FINAM to providing a user-friendly and adaptable framework for the environmental modeling community.

## 3 Coupling examples

In the following, we describe three workflows to demonstrate different features and fundamental concepts of FINAM. It should be noted that the focus is not on the scientific outcome of these coupling examples but rather on the technical realization.

### 3.1 Bidirectional toy model

FINAM allows to create circular couplings, enabling compositions where the output of one component serves as the input for another, and vice versa. There, iterative data exchange allows the development of complex and interconnected networks of components. This section presents this concept and provides an illustrative example of the bidirectional coupling features in FINAM using simple toy models.

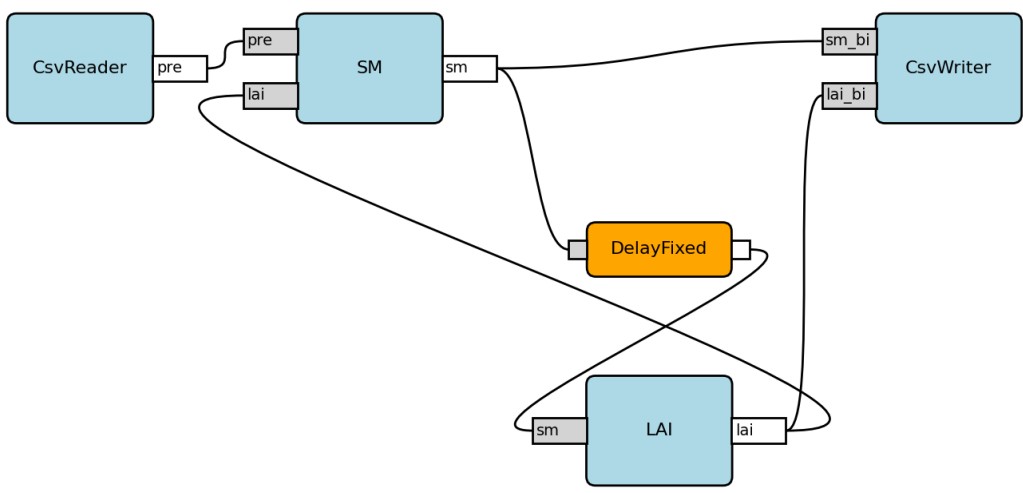

**Figure 12.** FINAM coupling diagram of a bidirectional model between LAI and SM.

We couple a toy model that simulates the leaf area index (LAI) of the plant canopy with a toy model that calculates soil moisture (SM) based on precipitation data. The toy models represent simplified first-order effects, where LAI decreases for a dry soil, and new plant biomass can only be created with sufficient soil moisture. For soil moisture, it is a simple water balance with precipitation as a source term and transpiration, represented as a linear function of LAI, as a sink term. These models are set up purely for demonstration purposes and the coupling is illustrated in Fig. 12. To demonstrate why a bidirectional coupling is beneficial in this case, a second scenario was built, where SM is calculated using a fixed LAI value for the entire simulation period (see Fig. A1).

The two toy models are defined by the following equations for SM (1) and LAI (2):

$$\mathrm{sm}(t) = \mathrm{sm}(t-1) + B \cdot \mathrm{pre}(t) - C \cdot \mathrm{lai}(t) \tag{1}$$

$$\mathrm{lai}(t) = A \cdot \mathrm{lai}(t-1) + (1-A) \cdot f(\mathrm{sm}(t)), \tag{2}$$

where $t$ is the time step index, $\mathrm{sm}[-]$ is the soil moisture, $\mathrm{lai}[-]$ is the leaf area index, $\mathrm{pre}\,[\mathrm{mm/d}]$ the precipitation, $A$ is a parameter to account for the relationship between LAI and SM, $B[-]$ and $C[-]$ are parameters to control the impact of precipitation and vegetation characteristics on the evolution of soil moisture. The term $f(\mathrm{sm}(t))$ represents the effect of soil moisture on LAI as a piecewise linear function. For a completely dry soil (sm is zero), it is zero, increasing to a maximum of five at sm of 0.6, and decreasing to three for completely saturated soil. Soil moisture is bounded to be between zero and one.

To enable a bidirectional coupling, we use a time delay adapter provided by FINAM as seen in Fig. 12. Since the components implement the equations (1) and (2), where both variables need the other already calculated for the current time step, we need to provide one model with past data to break the dependency cycle. We offset the SM input of the LAI component by one time step, which means that it uses the soil moisture of the previous day. Integrating the `DelayFixed` adapter replaces $sm(t)$ with $sm(t-1)$ in the coupled equation (2), thus delaying the effect of changes by one time-step. A five-year precipitation time series (1989 - 1993) was taken from the test domain of the mesoscale hydrological model (mHM) (Samaniego et al., 2010; Kumar et al., 2013), by extracting data from a single coordinate in space. This data was stored as a CSV file for reading by the FINAM `CsvReader` component.

The results of both models (unidirectional and bidirectional) are shown in Fig. 13. The upper panel shows the precipitation data used as input for the SM component in both models, while the lower panel showcases the two model results as time series of LAI (dashed) and SM (solid).

A key observation is the deviation from the bidirectional model (blue) and the unidirectional model (orange), especially in 1992. A dry period results in a dryer soil that effectively reduces LAI. This then results in less transpiration and a steeper increase in SM afterwards compared to the constant LAI case. This reflects the bidirectional interaction of the two toy models. One should note that the cyclic coupling of the two components only needed two lines of code in the composition script.

FINAM manages the complexities of time-stepping, data exchange, and synchronization, allowing users to focus on model development rather than integration logistics. Implementing bidirectional couplings is notably straightforward due to the modular design of the framework. In comparison to unidirectional coupling, where the data flow is one-way only and feedback loops are ignored, the difference in implementing bidirectional coupling is minimal. Adjusting the data flow from an LAI generator to the LAI component, for instance, involves only a few additional lines of code.

## 3.2 Coupling complex wrapped models

One major application of FINAM is the coupling of complex models to answer scientific research questions where a single model alone reaches its limits in terms of system boundaries or implemented processes. To demonstrate the applicability of FINAM for such tasks, in this section we present a unidirectional coupling of the systemic soil model BODIUM (König et al., 2023) and the component transport process implemented in OpenGeoSys (OGS) (Kolditz et al., 2012). BODIUM simulates the most important processes in soil and at the plant-soil interface on the field scale, including plant growth, and is developed for agricultural systems. The lower boundary is the rooting layer or a few centimeters below. This is the spatial boundary where a coupling to OGS is of interest. OGS is an open-source simulation software for thermo-hydro-mechanical-chemical (THMC) processes in porous and fractured media. The partial differential equations used for modeling are solved numerically using the finite element method.

Reducing the pollution of water bodies with nitrate from agricultural sources is an important challenge. To understand the fate and transport of nitrate from its application as a fertilizer to its entry into groundwater and rivers, model simulations are a powerful tool. In this coupling example, we use BODIUM to simulate two hypothetical agricultural fields in different locations managed with a winterwheat monoculture and different levels of nitrogen fertilization (170 kg/ha/a and 340 kg/ha/a,

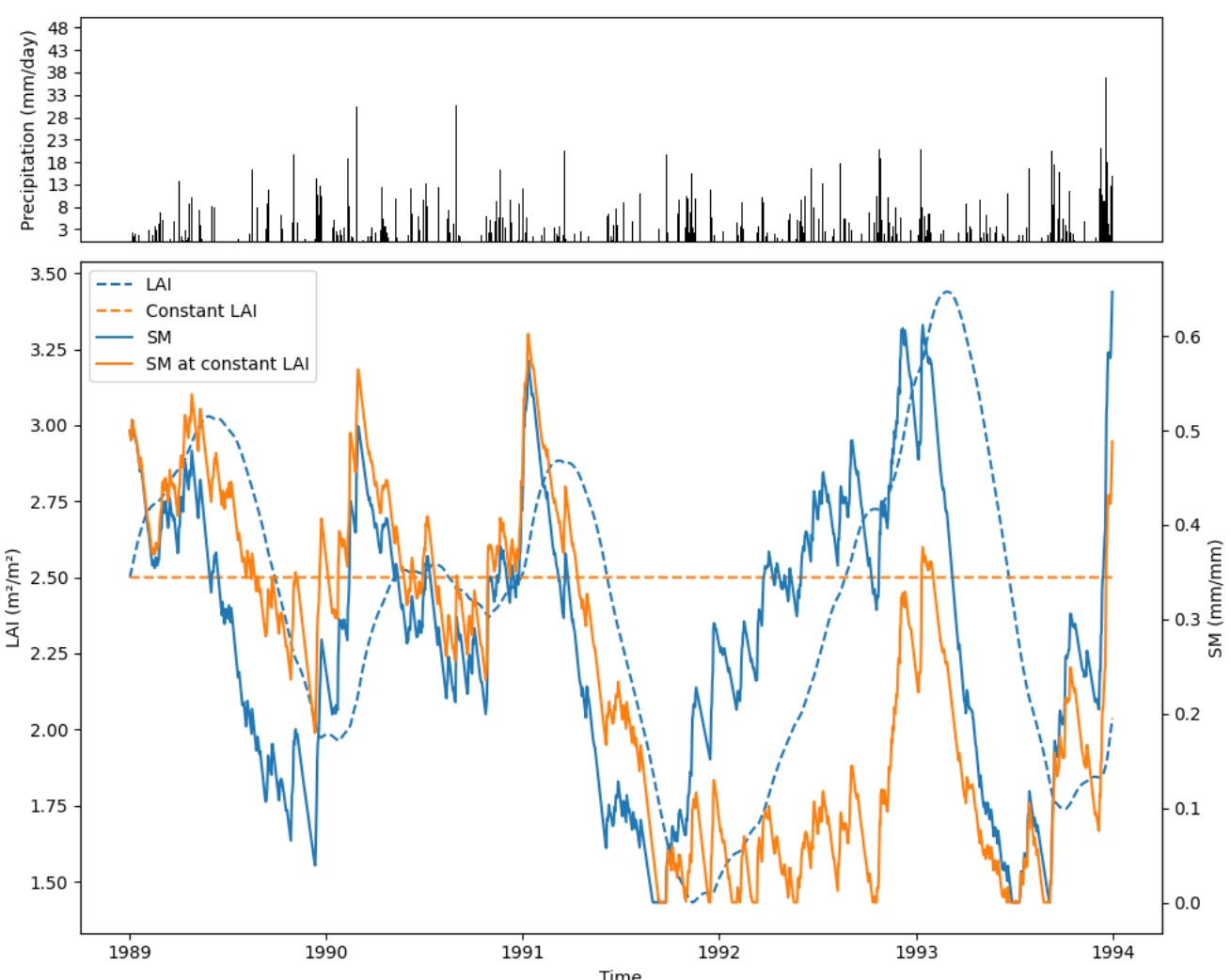

**Figure 13.** Precipitation (top figure), LAI (dashed lines), and SM (solid lines) time series for a FINAM uni- (orange) and bi- (blue) directional model coupling (bottom figure). Ticks mark the beginning of the year.

respectively). The simulated nitrate leachate from each time step is passed through FINAM to OGS, where the transport within groundwater and to a nearby river is simulated.

BODIUM and OGS operate on the same temporal scale (daily time step) but on different spatial scales. While BODIUM is a 1D model simulating on the field scale, OGS simulates on irregular grids on the catchment scale. However, the exchange variable in this specific coupling example is given in mass per area and thus is independent of the exact spatial distribution. Thus, we can apply a simple linear regridding within FINAM to overcome the spatial differences between the coupled models with BODIUM simulating two different instances for the two agricultural field sites. The basic diagram in Fig. 14 shows

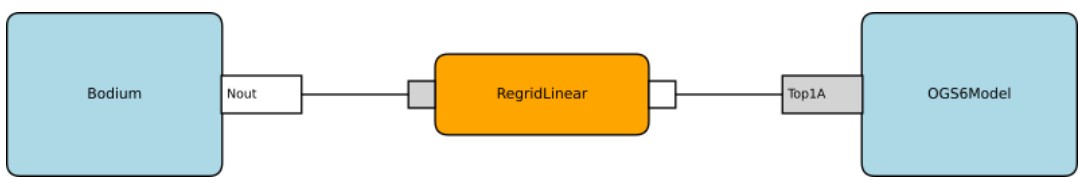

**Figure 14.** FINAM coupling diagram of BODIUM and OGS

the components involved (colored blue) and the linear regridding adapter (colored orange). The connection lines between the components and adapters indicate data exchange controlled by FINAM, from white outputs to gray inputs.

The surface of the model domain with $x$-extent of 1830 m, $y$-extent of 830 m, and $z$-extent of 48 m is given by a topography extracted from SRTM data. In the bottom panel of Fig. 15, the domain is exaggerated five times and the two agriculture sites are delineated by the two stripes, which are moved up a little for visualization purposes. In the upper panel of Fig. 15 the nitrate leachate is plotted over time.

For each agricultural site, a different BODIUM instance simulates nitrate leachate, which is passed to OGS at the specific field locations via the FINAM linear regridding adapter. At all other parts of the simulation domain, nitrate leachate passed to OGS is set to zero (assuming neglectable nitrate leaching in non-agricultural fields). The amount of nitrate leached into groundwater depends on the time of fertilizer application, precipitation, and nitrogen uptake by plants, resulting in temporal peaks of nitrate passing from BODIUM to OGS (top panel of Fig. 15). After the nitrate has reached the subsurface, it is transported along the groundwater flow field. The subsurface nitrate distribution is shown after 264 days in the left part and after 680 days in the right part of Fig. 15. In order to create a more dynamic behavior, both fields have slightly different time series of nitrate leach.

This example demonstrates how established complex models can be easily coupled via FINAM. However, note that both models had to be prepared for model coupling with existing python bindings and a FINAM wrapper, i.e. the interface shown in Fig. 4 has to be implemented. In addition, to apply this coupling for advanced research questions, further extensions of the coupling would be of interest, such as a bidirectional coupling by passing the hydraulic head from OGS to BODIUM.

## 3.3 Spatio-temporal regridding

Spatio-temporal regridding is a fundamental concept in the field of spatial data analysis and processing. Thus, regridding algorithms are a big part of the mentioned coupling frameworks and libraries ESMF, OASIS or YAC. It involves the transformation of data from one spatio-temporal grid or coordinate system to another. In FINAM the coordinate reference system (CRS) is part of the grid definition to account for Cartesian and spherical coordinates. Coordinate transforms are performed automatically when grids have different CRSs.

FINAM provides built-in regridding adapters for linear and nearest-neighbor interpolation designed to easily transform data from one model to another. For advanced regridding operations, we provide a dedicated package `finam-regrid`, which wraps the regridding algorithms of ESMF (Collins et al., 2005), covering a wide range of regridding methods, grid types, and

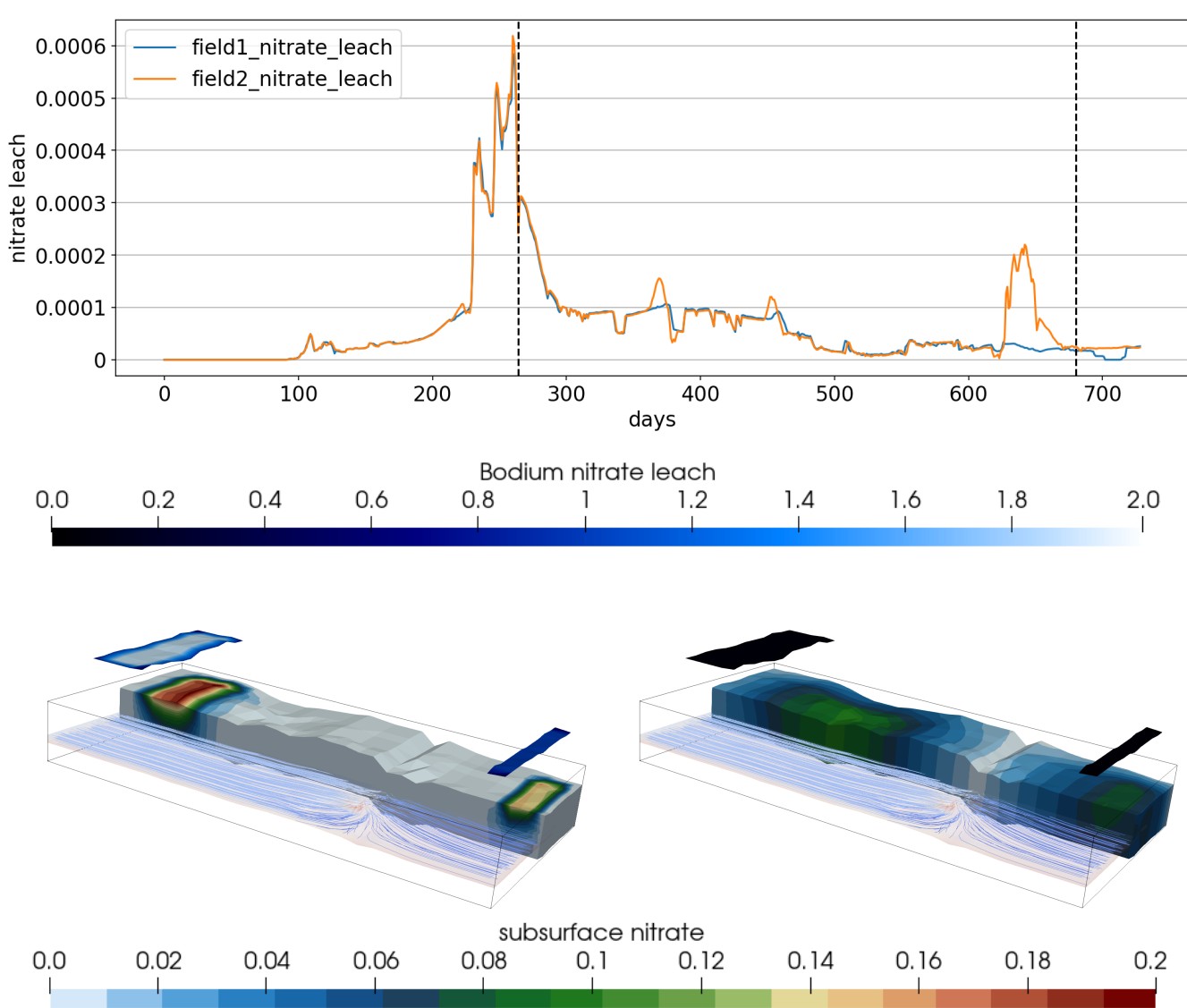

**Figure 15.** Top panel: nitrate leachate computed by BODIUM for the two different fields (field1 and field2); bottom: subsurface domain outline and (for visualization purposes translated as 'clouds') boundary patches above coloured by BODIUM nitrate leachate; in the background (exaggerated 5 times) nitrate concentration computed by OGS: left: after 264 days, right: after 680 days, in the front groundwater flow paths coloured by velocity

```
import finam as fm
import finam_regrid as fm_rg
...
# Adapters
method = fm_rg.RegridMethod.CONSERVE
regrid = fm_rg.Regrid(regrid_method=method)
mean = fm.adapters.AvgOverTime(step=0)
...
# Connections
reader["pre"] >> regrid >> mean >> writer["pre"]
```

**Figure 16.** FINAM composition excerpt to regrid and average precipitation over time. The reader and writer modules specify the spatial and temporal resolutions of the source and target precipitation data. The writer is pre-configured with a regular mesh and a writing frequency of 30 days. This implies that the regrid and mean adapters do not need further information about the data specification, as it is determined from the connected components. The regridding method is set to CONSERVE in order to preserve the total amount of precipitation and the averaging adapter is configured to interpret the data to be valid for the time span right before the time stamp (step=0).

coordinate systems. For details, we refer to the documentation of finam-regrid[9]. These adapters are designed to translate
data between different grid formats - structured or unstructured - and adjust resolutions to ensure compatibility between models. Incorporating regridding capabilities directly within the framework significantly reduces the workload on model developers and users. Performing as a dynamic adapter, it automatically detects the specifications of the source and target grids and derives the required transformation for differing CRSs. The regridding method to use needs to be specified by the user and is depending on the exchanged data. Despite the underlying complexity of this task, we have engineered the tool to be user-friendly, ensuring
that its advanced capabilities are accessible without the need for detailed technical knowledge of regridding.

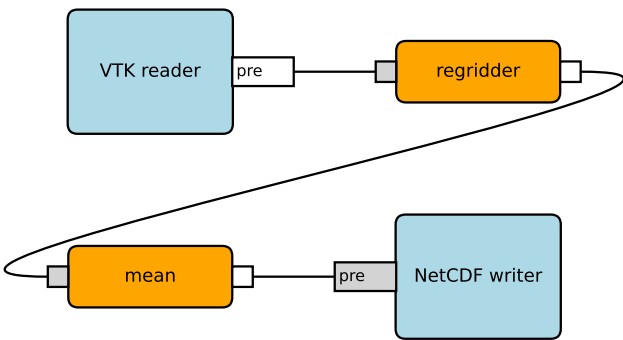

**Figure 17.** FINAM coupling diagram of the spatio-temporal regridding of precipitation.

---

[9]https://finam.pages.ufz.de/finam-regrid

In this example, we illustrate the spatio-temporal regridding capabilities of FINAM by converting daily precipitation data from an unstructured grid, with a cell edge length of approximately 0.5 km, to a 30-day mean precipitation on a regular grid with 1 km cells, covering an area of five by four kilometers. To simplify the example, we utilize the finam-vtk package to read the unstructured data time series stored in the PVD format and then write the data using the finam-netcdf module. However, this workflow could also be integrated into a larger system in which meteorological data from an atmospheric model serves as input for a crop yield estimator that operates on monthly data.

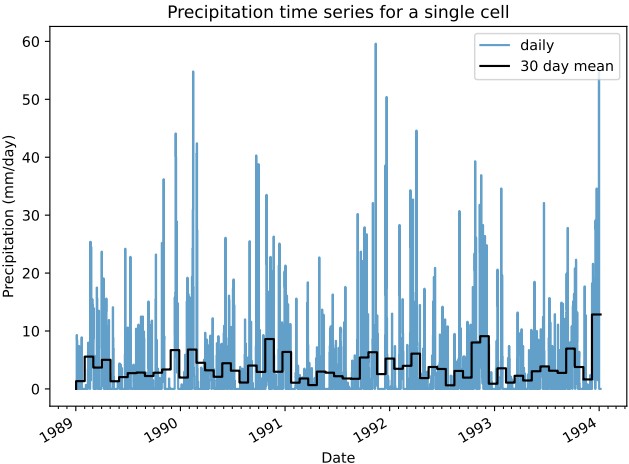

**Figure 18.** Temporal aggregation of precipitation data for a single grid cell. The five year input time series of daily data is shown in blue and the 30-day rolling average is shown in black.

Fig. 16 shows the important parts of the coupling setup script, which shows the ease of configuring the data flow, while Fig. 17 shows the overall coupling scheme. We employ a five-year time series of daily data and apply a rolling mean with a 30-day window after regridding the data to the structured target grid. Fig. 18 shows the averaging of an extracted time series from the top right grid cell. The time averaging adapter is capable of converting the data on the fly with almost no configuration. Input and output time stepping is purely derived from the connected components, where the input precipitation is pushed daily by the reader component and the writer will request data every 30 days. This highlights the ease of creating data streams, where users do not need to worry about time-stepping compatibilities. Furthermore, Fig. 19 shows the results of the regridding for a single day on the unstructured grid and the average of one month on the target grid, which includes that day. To ensure that the total amount of precipitation is preserved, in this example the conservative regridding implementation of ESMF was used. Again, the adapter for spatial regridding will determine its configuration entirely from the data specifications of the connected components, which avoids sources of error.

This workflow demonstrates the ease of creating data workflows that connect components operating on different spatio-temporal scales. By regridding daily precipitation data from an unstructured grid to a regular grid and aggregating it into

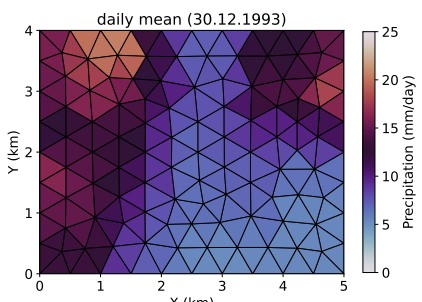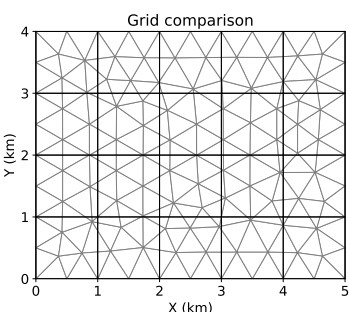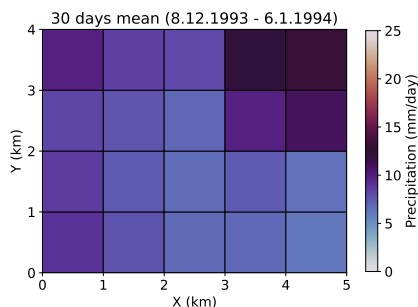

**Figure 19.** This figure illustrates the regridding process of precipitation data based on ESMF wrapped by `finam-regrid`. The left panel shows the input unstructured grid for a single day, highlighting the original distribution and variability of the data on an irregular grid. The middle panel overlays the unstructured and resulting structured grids, without displaying data, to emphasize the differences in grid configurations and cell outlines. The right panel presents the resulting structured grid with time-averaged precipitation data over 30 days, including the single day from the left panel, demonstrating the transformation and smoothing effect achieved through regridding.

30-day means, it showcases the integration of various data formats and the seamless data flow configuration, highlighting the potential for coupling diverse data-driven models.

## 4 Discussion

In the development of the model coupling framework FINAM, our primary goal was to address the significant challenges inherent in coupling models written in diverse programming languages, operating on different time steps and grid systems. The selection of Python as the foundational language for this coupler was predicated on several of its intrinsic qualities that make it exceptionally suited for such a complex task.

Python's interoperability is enhanced by a variety of libraries that facilitate the development of wrappers for integrating models across different native programming languages like C, C++, Fortran, or Rust. Additionally, Python's extensive ecosystem and library support simplify the process of model creation and data processing, backed by a strong, global community of developers and researchers. This supportive environment, combined with the cross-platform nature of Python and the ability to rapidly prototype it, creates an optimal setting for the development and fast and easy testing of complex model couplings.

One fundamental idea of FINAM is the assumption that temporal models inherently contain a time loop within their code structure. The coupler makes use of this characteristic by requiring that each model needs to be able to perform a single time iteration. This should be controlled by interface routines provided in Python that also provide access to the internal states between these iterations. Such an approach allows for the seamless implementation of a FINAM component to wrap around any model, facilitating its integration into the coupling framework without necessitating direct further modifications to the model's source code, when Python-bindings are available.

Compared against a pure pull-based approach, where models are only executed if data is requested by downstream components, FINAM's scheduling provides several advantages. Firstly, a pure pull-based approach requires a single component that is the end point of the coupling, which drives the complete chain via pull. With FINAM's approach, this is not required and a coupling setup can have an arbitrary number of end points. Secondly, in a pull-based approach, it is not guaranteed that all components run over the entire simulation time frame. In FINAM, this is guaranteed. Finally, the hybrid approach allows for push-based components that react to new data becoming available.

In contrast, solutions based on coupling libraries like OASIS often involve the integration of routines in their supported languages directly into the model source code, enabling data exchange during runtime (Hanke et al., 2016; Craig et al., 2017). However, this method may impose a significant burden on modelers, who must ensure data compatibility with respect to units, grid definitions, and time references, potentially necessitating extensive modifications and conditional extensions to the original model code. FINAM distinguishes itself by offering utilities to validate and process data on the fly, thereby simplifying the coupling process and minimizing the need for direct alterations to model code. However, as described in section 2.2, some refactoring might be necessary.

To further emphasize the practical advantages of the FINAM framework, it is essential to highlight the usability of the adapters provided within the system. These adapters enable smooth data exchange between models with varying data requirements and formats, without the need for explicit configuration. As an example, regridding adapters automatically determine their required transformation from the connected source and target components. Regridding is not required if grids are compatible, in the sense that only trivial transformations like axis flipping or transposing are needed.

In addition, FINAM includes a variety of readers and writers designed to handle multiple file formats, such as NetCDF, VTK, or CSV. This versatility allows researchers to integrate models from different domains without the need for time-consuming conversions or extensive preprocessing. All data exchanged in FINAM is wrapped by `pint` to enable automatic unit conversion and checking. This means that incompatible inputs and outputs can never be connected by mistake, and, if required, a unit conversion is performed on the fly.

However, FINAM still has several limitations and potential for improvements and extensions. One of the main challenges FINAM is currently facing is the issue of parallelization. Particularly, the integration of MPI (Message Passing Interface) for parallel computing, to distribute models to different CPU cores, is work in progress. There are also Python-based parallelization approaches (multiprocessing[10], joblib[11], dask[12], etc.) that could be used in the future to run independent parts of the composition in parallel. Other frameworks like YAC that are built on top of MPI support this out of the box but follow a different approach of model coupling. Additionally, making a model ready for FINAM can require a significant investment of time and resources if the source code is not well structured. Although Python facilitates rapid prototyping and development, preparing a model to meet FINAM's requirements can be a demanding process, particularly for legacy systems or highly complex models, depending on their internal structure. If the code base is well written and its functionality is encapsulated in logical units, such

---

[10]https://docs.python.org/3/library/multiprocessing.html

[11]https://joblib.readthedocs.io

[12]https://www.dask.org/

as executing the time loop as a separate subroutine, the Python interface is reasonably easy to implement. This is for example the case, when a model already follows the Basic Model Interface (BMI).

Another technical limitation arises from the nature of Python itself. Preventing unnecessary data copying within the FINAM framework can become complicated due to different internal data representations in different models. By default, numpy masked arrays are used to exchange data, but these arrays could hold copies instead of views to the underlying data. Efficiently managing data without unnecessary copying is crucial for maintaining performance, especially when dealing with large datasets or high-frequency data exchanges. This challenge underscores the need for ongoing development within the FINAM framework to enhance its efficiency and reduce overhead, ensuring that it remains a viable solution for complex model coupling scenarios.

We implemented a set of benchmarks in the test suite of the FINAM repository to track the computational overhead of the framework. Using Fromind and mHM, comparisons between native model runs and model executions via FINAM without any data exchange have shown negligible differences in runtime. When data arrays were exchanged by daily time steps, the tests have shown a 5% overhead for one exchanged variable (soil water from the standard test domain of mHM) compared to standalone runs, which we think is reasonable with respect to our target use cases.

The last area to mention is the handling of metadata and configuration. Although the basic infrastructure to track metadata from components has been implemented, there is currently no best-practice guide on how to utilize this feature effectively. Establishing such guidelines would help standardize metadata usage, making it easier for users to follow the *FAIR* data principles (Barker et al., 2022). Together with metadata, there is the lack of a unified configuration system for compositions. This can make it challenging to reuse compositions in complex computational workflows, potentially hindering the efficiency and reproducibility of simulations. Developing a unified configuration approach would streamline the setup process and simplify the sharing and replication of complex model compositions.

A unique feature of FINAM is its support for bidirectional coupling by temporal delaying circular input/output connections. While other couplers do this implicitly based on the code position of their getters and setters, FINAM provides adapters to explicitly control the time offset of the exchanged data. Using this method, we do not require models to save and reset states, a process that can be overly demanding for many models. The gained flexibility is further enhanced by FINAM's approach to time representation. By assuming that valid time spans for data are defined by timestamps, with the current timestamp indicating the end of the reference period that started from the previous timestamp, FINAM standardizes time representation across models. This method clarifies the time frame for *extensive* variables, like total precipitation of a day, and provides a precise reference for *intensive* variables, such as air temperature for a specific point in time, without requiring direct modifications to the handling of time of the models.

Furthermore, FINAM includes time adapters designed to bridge models with differing time steps and spans through techniques such as integration, summation, and temporal disaggregation. This capability not only enhances the framework's flexibility, but also significantly reduces the complexity involved in model integration, allowing researchers to focus on the substantial aspects of their work without being encumbered by technical incompatibilities.

## 5 Summary and Outlook

In this study, we present FINAM, short for "FINAM Is Not A Model", which is a Python based coupling framework designed to
460 connect models written in diverse programming languages. Models are wrapped in components with a well-specified interface
that facilitates the exchange of data between them. Additionally, FINAM provides a set of tools to process data that includes
functionality such as regridding and unit conversion, but also reading and writing different file formats (such as NetCDF, VTK,
and CSV). FINAM handles data compatibility checks, unit conversion, and component scheduling, which makes the model
coupling process less error-prone. It provides a unique and intuitive mechanism to link components, which makes it easy to
465 set up a coupled model. We presented three examples that highlight different features of FINAM. The first example consisted
of two toy models, simulating soil moisture and LAI with cyclic dependencies on each other to demonstrate the capabilities of
FINAM of handling dynamic feedbacks between components. The second example illustrates the usage of FINAM to couple
the separately developed models BODIUM and OGS to simulate the transport of nitrate leachate in groundwater. In a third
workflow, we demonstrated important features to spatio-temporal regrid data within a FINAM coupling. To do so, we remapped
daily precipitation data given on an unstructured grid to a coarser structured grid and applied a rolling average of 30 days.

In summary, FINAM provides a flexible and user-oriented approach to model coupling, leveraging Python's versatility
to integrate a wide range of models, such as environmental models, including ecological models for animal populations,
individual-based forest models, field-scale crop models, economical models, and hydrological models, but also surrogate models or machine learning models. By minimizing the need for direct modifications to the model code and offering innovative
solutions to handle time representation and data compatibility, FINAM represents a significant advancement in the field of
computational modeling.

*Code availability.* The code of `FINAM` is developed at https://git.ufz.de/FINAM/finam and available via Zenodo at https://doi.org/10.5281/
zenodo.7602944. It is distributed under the GNU LGPL v3.0 license. The documentation, which includes a quick start guide, a more in-
depth hand book, and a complete overview of the API, can be accessed via https://finam.pages.ufz.de. All mentioned expansion modules and
480 examples are hosted under https://git.ufz.de/FINAM.

*Author contributions.* Martin Lange and Sebastian Müller are the main developers of FINAM and contributed both equally to all sections
of the text. Sara König and Thomas Fischer wrote the section about coupling complex wrapped models and helped proofreading the text.
Jeisson Javier Leal Rojas wrote the section on the bidirectional toy model. Stephan Thober wrote the introduction and the summary and
supervised the project. Matthias Kelbling helped with implementation details and helped to improved the text.

*Competing interests.* The authors declare that they have no conflict of interest.

*Acknowledgements.* Sara König was funded by the German Federal Ministry of Education and Research (BMBF) in the framework of the funding measure 'Soil as a Sustainable Resource for the Bioeconomy—BonaRes', project 'BonaRes (Module B): BonaRes Centre for Soil Research'. This work is a contribution to the LandTrans simulator initiative at the Helmholtz Centre for Environmental Research - UFZ.

## Appendix A: Unidirectional coupling of LAI and SM

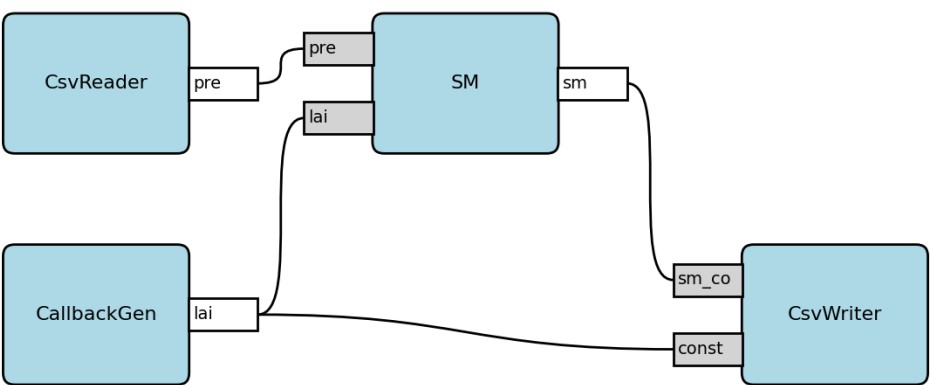

**Figure A1.** FINAM coupling diagram of a unidirectional model between LAI and SM.

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
