# Peer review of "FINAM - is not a model (v1.0): a new Python-based model coupling framework"

_Geoscientific Model Development, 2024_

## Author Response (AR1)

**Rebuttal**

A point-by-point response to the reviews including a list of all relevant changes made in the manuscript "FINAM - is not a model (v1.0): a new Python-based model coupling framework".

Sebastian Müller Martin Lange Thomas Fischer Sara König Matthias Kelbling Jeisson Javier Leal Rojas Stephan Thober

December 19, 2024

**Chapter 1**

**Reviewer:** Moritz Hanke**

We thank you, Moritz, for your detailed review of our work and the constructive comments you made. These help to further improve the quality of our manuscript.

**1.1** General remarks:**

**Original comment:**

In the introduction, as the motivation for the development of FINAM, common coupling strategies and examples for their implementations are presented. Additionally, their disadvantages are listed. However, what the authors fail to do is to start by giving an overview on the main use-case for which FINAM is currently built for and the potential target users, which is different from compute intensive and highly-parallel weather and climate model, which OASIS and YAC are used for. If this had been established first in the introduction, the characteristics of the various coupling solution could be listed.

**Response:**

Thank you for this suggestion. We agree that it is important to clarify the target use cases of FINAM early on in the manuscript.

FINAM is targeted for general environmental models that might be gridbased or not. Examples include ecological models for animal population, individualbased forest models, field-scale crop models, economical models, but also hydrologic models (see mHM, OGS). FINAM has the goal to enable these models to exchange data in a concise and flexible way. These application do typically not require HPC resources unlike the mentioned earth system models.

We will include a more detailed explanation early in the introduction, emphasizing that FINAM is primarily designed for environmental modeling rather than highly parallel, computationally intensive weather and climate models.

**Manuscript changes:**

We modified the abstract to better frame the target use cases and the general focus of FINAM: "Although established coupling solutions such as YAC, ESMF, or OASIS focus on highly parallel workflows, complex data processing, and regridding, FINAM prioritizes usability and flexibility, allowing users to focus on scientific exploration of coupling scenarios rather than technical complexities. FINAM emphasizes ease of use for end-users to create, run, and modify model couplings as well as for model developers to create and maintain components for their models. The framework is particularly suited for applications where rapid prototyping and flexible model extensions are desired. It is primarily targeting environmental models, including ecological models for animal populations, individual-based forest models, field-scale crop models, economical models, as well as hydrologic models."

**Original comment:**

Based in this, the need for the newly developed FINAM could be explained. For example library based coupling solutions like OASIS and YAC are used for highly-parallel model with long histories and large code basis. A restructuring of these models for coupling framework would be very difficult and is often not supported by the associated communities. They usually have fixed time step length, and therefore the support for variable time step length is not a requirement. Because this is an important information for many readers, it should be stated somewhere at the beginning of the paper, that currently FINAM runs in a single process and does not support MPI.

**Response:**

As stated above, FINAM is not primarily targeted for HPC resources. FI-NAM does not support MPI currently, which is a fundamental necessity for other coupling frameworks like OASIS, YAC or ESMF. This limitation of FINAM is also highlighted in Section 4 (L362).

The need to mediate between models with different time steps was indeed a fundamental requirement, since the initially selected models cover a wide range (see also reply above) and typically operate on different time-scales.

Manuscript changes:

No changes.

**Original comment:**

For many coupling setups the performance penalty of the coupling is an important factor, which is why publications related to coupling solutions often include respective measurements. However, in your use-case this does not seem to apply. This could be a useful information to the reader.

**Response:**

FINAM's primary use case does not require significant performance optimization, unlike other coupling frameworks like YAC or OASIS. We will highlight this in the revised manuscript to manage reader expectations on performance measurements. In addition to that, we have benchmarks in our testing suites in the FINAM repository. We optimized FINAM to have as little overhead as possible and tested this by comparing native model runs with runs controlled by FINAM. We will add information on that to the paper. We tested in-house models like Formind and mHM for the execution overhead introduced by FINAM. The plain running overhead, when no data is exchanged is negligible. When data is exchanged, it needs to be copied due to the impossibility to require all models to use the same memory layout internally and thus the overhead increases in this case. Tests with Formind and mHM have shown a 5% overhead for one exchanged variable (soil water) compared to standalone runs. The included benchmarks in the FINAM repository also indicate that the biggest share of the run-time overhead comes from data copying, which is to be expected and is also the case for other couplers that build on MPI for example.

**Manuscript changes:**

We added the following paragraph to the discussion: "We implemented a set of benchmarks in the test suite of the FINAM repository to track the computational overhead of the framework. Using Fromind and mHM, comparisons between native model runs and model executions via FINAM without any data exchange have shown negligible differences in runtime. When data arrays were exchanged by daily time steps, the tests have shown a 5% overhead for one exchanged variable (soil water from the standard test domain of mHM) compared to standalone runs, which we think is reasonable with respect to our target use cases."

**Original comment:**

Other coupling solutions you could take into account for your state-of-the-art analysis are:

- bmi (https://bmi-spec.readthedocs.io/en/latest/)
- OpenPALM (https://www.cerfacs.fr/globc/PALM\_WEB/index.html)
- preCICE (https://precice.org/)

**Response:**

We appreciate the suggestion to consider bmi, OpenPALM, and preCICE. We will include references to these tools in our state-of-the-art analysis to provide a more comprehensive comparison of coupling solutions.

The bmi specifications are very well suited to build a FINAM component from a model that implements a Basic Model Interface (BMI). In contrast to FINAM, BMI does not provide routines to handle the scheduling and the exchanged data also needs to be retrieved and forwarded manually during the execution. The BMI basically specifies what a FINAM component would require from the Python bindings of a model.

While OpenPALM excels in managing complex, large-scale simulations across diverse models, its complexity and setup requirements make it less suitable for rapid prototyping in Python. FINAM, with its Python-centric design and emphasis on simplicity, offers a more accessible and efficient solution for quickly developing and testing coupled models which better fits our target user base.

The same is true for preCICE which is an open-source coupling library specialized in partitioned multi-physics simulations like fluid-structure interaction and heat transfer and therefore very domain specific.

Since these frameworks are built to incorporate different models, one could implement interfaces in form of a specific Component to be able to connect existing couplings with a FINAM composition. We already talked about this internally in the past as one could see here: https://git.ufz.de/FINAM/finam/-/issues/106

**Manuscript changes:**

We added the following paragraph to the introduction: "There are several other domain-specific coupling solutions like preCICE (Chourdakis et al. 2022),

which is an open source coupling library specialized in partitioned multiphysics simulations such as fluid-structure interaction and heat transfer, or OpenPALM (Buis et al. 2006) which is specialized on complex systems and highly parallelized computations. For completeness, we also mention the Basic Model Interface (BMI) (Hutton et al. 2020), which is not itself a coupler but rather a standardized, language-agnostic interface specification that models can implement to simplify interoperability and coupling."

**1.2** Unrelated to the Paper:**

**Original comment:**

You use the "datetime" class from Python, which uses the Gregorian calendar. Can you exclude the possibility that FINAM users may require different calendars at some point?

**Response:**

The current implementation relies on the Gregorian calendar, which is compatible with the majority of environmental models. We currently do not expect users to rely on calendars that cannot be easily converted to the Gregorian (e.g., a 360 day calendar). Currently, component developers need to convert the dates internally and provide this as the time-stamp in FINAM. We will expand on the use of calendars in the manuscript. One idea could be to use the implementations of cftime (https://github.com/Unidata/cftime), to be able to use a wider set of calendars, but this still would only work if all components use compatible calendars.

**Manuscript changes:**

We added a short paragraph to the "Data and metadata" section to clarify this: "Since we use the built-in datetime module of Python, we require all models to provide their temporal data on a Gregorian calendar which is a reasonable restriction for environmental models."

**1.3** Specific Comments:**

**1.3.1 Abstract:**

**Original comment:**

You should probably include a hint on the main use case for FINAM, because it might be different for other coupling solutions like ESMF or OASIS. Because this is an important information for potential readers of this paper.

**Response:**

As stated above, we state the main goal of FINAM more clearly in the revised manuscript.

**Manuscript changes:**

See above.

**Original comment:**

• L3: "independently developed source codes" The development of two models can depend on each other (one does not work without the input from the other) and can even have a partially shared code base (e.g., ICON atmosphere and ICON ocean) but still require coupling. Maybe try to find a better wording.

**Response:**

We will rephrase "independently developed source codes" to clarify that we mean models that were developed as stand alone tools in the first place, which was a fundamental motivation for FINAM.

**Manuscript changes:**

We rephrased the second sentence of the abstract: "FINAM is designed to facilitate the coupling of models that were developed as stand-alone tools in the first place, and to enable seamless model extensions by wrapping existing models into components with well-specified interfaces."

**Original comment:**

• L13: add reference for OpenGeoSys and Bodium

**Response:**

References for OpenGeoSys and Bodium are present later in the manuscript and we don't want to include references in the abstract.

Manuscript changes:

None.

**1.3.2 Introduction:**

**Original comment:**

Add description of targeted use case and the characteristics of applications FINAM is supposed to be used for. This is only slightly hinted at by "environmental models". A comparison of the characteristics of these models compared to the ones used in weather and climate models (for which ESMF, OASIS, and YAC are used) would greatly improve the understanding of the evaluation done in the introduction. In case you are interested, there are publications by Sophie Valcke in which she compares the coupling library and framework approaches.

**Response:**

Again, we will improve the description of FINAM's target use case and expand on the differences between environmental models and weather/climate models. Thanks for pointing out the paper of Sophie Valcke1. As the title says, this is again a comparison of couplers for earth system models which is not in the targeted scope of FINAM.

**Manuscript changes: See above.**

1. https://gmd.copernicus.org/articles/5/1589/2012/

**L21:**

Original comment: "independently developed models" (see above) Response: See our response above.

**L21:**

**Original comment:**

"three main approaches to exchange data" You could also add the option to include one model into another by merging the source code or through a plugin-mechanism. In this case the data is exchanged directly through routine arguments between the models.

**Response:**

We will add the approach of merging source codes or using plugin mechanisms as another strategy for coupling models.

**Manuscript changes:**

"A fourth option, which we will not elaborate on further, are custom solutions like merging code bases of different models or rewriting these from scratch."

**L25:**

Original comment: "infeasible" maybe "impractical" Response: We will rephrase "infeasible" to "impractical". Manuscript changes: We rephrased "infeasible" to "impractical".

**L28:**

Original comment:

"infeasible" duplication

**Response:**

We will replace this second occurrence of "infeasible" with another suitable word.

Manuscript changes:

We replaced the first "infeasible" and kept the second.

**L35:**

**Original comment:**

"Shrestha et al., 2014" The work in this paper used OASIS3 and not OASIS3-MCT, which makes a significant difference especially in respect to performance.

**Response:**

Thank you for bringing this to our attention, we will correct the reference to mention that OASIS3, not OASIS3-MCT, was used.

**Manuscript changes:**

We removed the reference to Shrestha et al., 2014 and only keept the first one for OASIS3-MCT.

**L38:**

**Original comment:**

"A disadvantage of these libraries [...]. This is an error-prone approach." That depends on the perspective of the evaluation and the experience of the users. So in your case, this may be an inadequate approach for coupling.

**Response:**

The experience from our user group is that past couplings were established in a file based manner, which means, the output of one model was converted to a meaningful input of a second model by hand. This process requires careful attention to numerous details, such as grid specifications, time step compatibility, units, and other metadata. Such a workflow is inherently error-prone due to the manual nature of the adjustments and the potential for mismatches or oversights.

Similarly, coupling methods that integrate directly into the codebase, such as those relying on getters and setters, can also inherit these error-prone characteristics. In these cases, developers must still manually ensure compatibility across models, explicitly define data exchange routines, and maintain consistency in metadata such as units ore reference time frames. These tasks, embedded within the source code, can be demanding and increase the maintenance burden, particularly when models evolve or are adapted for different contexts.

We acknowledge that the perception of error-proneness and the adequacy of coupling approaches vary depending on the users' experience and the specific use case. To address this, we will revise our statement to clarify that the challenges we describe primarily reflect feedback from our target user group and that it "can" be error-prone approach. For them, these traditional and library-based coupling approaches (whether file-based or code-integrated) can introduce significant risks of error and additional maintenance effort, especially when compared to more automated or metadata-aware solutions like we want to achieve with FINAM.

**Manuscript changes:**

We rephrased the last sentence to: "This can be an error-prone approach, especially for inexperienced users who want to focus on the scientific problem rather than the coupling implementation details."

**L40-41:**

**Original comment:**

"The maintenance of the data exchange..." That depends on the software design of a model. From my experience users of the uncoupled ICON atmosphere never have to deal with any coupling-related source code or configuration parameters.

**Response:**

We will rephrase the statement to reflect that this issue might depend on the software design of the specific models in question. This will also be much clearer once we refine the targeted use cases definition, since the users of FINAM will be the ones that write the coupling scripts and not only execute existing ones.

**Manuscript changes:**

We softened the statement to: "Depending on the software design, the maintenance of the data exchange calls in each model also may create additional work for model developers because they are not used in the "offline" model version."

**L49-51:**

**Original comment:**

"The disadvantage..." Even if my experience with ESMF is very limited, I do not agree with this statement. There is an example where ESMF was used for the coupling of ICON (see https://gmd.copernicus.org/articles/14/4843/2021/). Characteristics of ESMF that might make it unsuitable for your application is potentially its size and the complexity of the interface. Having the control over the development of the coupling software can also be a significant advantage.

**Response:**

We will clarify this point by highlighting that ESMF might be unsuitable for some applications due to size and complexity, while still acknowledging its successful use such as the ICON coupling mentioned. Again, rapid prototyping was a main motivation for FINAM which aligns with your statement that existing libraries may have another focus and require more coding to implement their required interfaces.

**Manuscript changes:**

We extended this sentence: "The disadvantage of this approach is that frameworks such as ESMF, while successfully used to couple independent codes without a complete rewrite in some large-scale applications (e.g., a coupling of the atmosphere model ICON and the coastal ocean model GETM (Bauer et al. 2021)), are generally designed to build model systems from the ground up. As a result, they may be less suitable for independently developed models with existing code bases, where significant restructuring could be required."

**L52-65:**

**Original comment:**

whole paragraph Based on my previous comments, this paragraph does not make it clear why there is a need for FINAM and how it is different from other solutions. It lists a couple of desired features that, depending on your point of view, might also be fulfilled by other software.

**Response:**

We will clarify the need for FINAM, particularly its distinction from other libraries in terms of features like bidirectional coupling and time management. We want to stress, that one aim was that coupling scripts can be written in a couple of minutes to enable modelers to tinker around with model configurations with a huge set of readily provided tools that are well documented.

**Manuscript changes:**

We rephrased the whole paragraph: "FINAM (short for "FINAM Is Not A Model") aims to fill this gap by prioritizing usability and ease of coupling over extreme computational performance. Our goal is to enable scientists to couple models with minimal effort, allowing them to comfortably experiment with model setups and focus on scientific exploration rather than technical complexities. FINAM allows for the coupling of independently developed codes and seamless model extensions by wrapping existing models into components with well-specified interfaces."

**L55-56:**

**Original comment:**

"FINAM is located in the middle between a coupling library and a coupling framework" I do not understand why FINAM is located between a coupling library and framework. What makes its approach significantly different from the framework approach of ESMF?

**Response:**

In order to prevent confusion about what the explicit distinction between a coupling library and a coupling framework is, we will remove this statement. For us, the difference between both is that a coupling library provides coupling routines that can be called from the model source-code directly, while a framework provides templates to implement components and handles the data-flow between these.

Consequently it would also make more sense to call FINAM a framework in this regard. We only wanted to highlight, that you can either wrap existing code via Python bindings (calling subroutines in the code-base by a wrapper) or implement components directly in a specified component classes (framework approach).

**Manuscript changes:**

We removed the statement.

**L75:**

**Original comment:**

"makes it easy for developers to wrap existing models" This is contradicted by L364-365 ("making a model ready for FINAM can require a significant investment of time and resources.". Especially for existing large code bases this can be an extensive task.

**Response:**

We will address the seeming contradiction by clarifying that while wrapping models for FINAM is relatively straightforward, significant effort may be required for models with large or complex code bases (e.g., Earth system models). We mean that it should be as easy as possible with the given restrictions.

**Manuscript changes:**

We rephrased this section: "This concept generally makes it straightforward for developers to wrap well-structured models (see Section 3.2 for details) and, once wrapped into a FINAM component, for users to set up and run compositions."

**L76-77:**

**Original comment:**

"Consequently, components can be developed in isolation without detailed knowledge of the potential coupling partner models" Depending on the complexity of a model, this can also be true for the coupling library approach. And actually is one of the ideas for a coupling library, unfortunately in practice this is often not the case.

**Response:**

Indeed, this should be true for any component based coupling solution and that is why we made the statement. Since the coupling of independently developed code was a motivation for FINAM we wanted to highlight that once they are FINAM ready, they can still be developed further independently.

Manuscript changes: No changes.

**L79-80:**

**Original comment:**

Here you list again coupling approaches including the one I proposed to be added to L21.

**Response:**

We will mention the coupling approaches proposed above for consistency. Manuscript changes:

We realized that the proposed option "merging code bases" is already in the text. Hence, no changes.

**L81-89:**

**Original comment:**

whole paragraph This paragraph compares FINAM to other approaches:

• FINAM does require very little framework-specific code

- Are you sure that in your examples, the same could not be said for ESMF?
  - \* Response: Using ESMF would require linking your code-base against ESMF and implementing an interface for ESMF specifically. What we mean by "generally useful functionality" is, that your time-loop is placed in a callable subroutine to make it accessible for Python bindings and this is not framework specific for FINAM.
- Looking at the code of the Hargreaves-Samani example, the amount of framework/library-specific code would be more or less the same, if a similar component would have been coupled using YAC.
  - \* Response: We know that other frameworks are able to do similar things. As stated, we wanted to use this example as a guiding light to explain the implementation details which we don't want to spare out.
- only minimal knowledge of the model is required
  - This is a very broad statement. (e.g. remapping between grids is handled by YAC automatically, but unit conversion and handling of varying time steps is not).
    - \* Response: We still think this point is true and didn't want to give the impression, that this is not true for any of the other solutions. See our overall response below.
  - btw. the selection of the regridding method depends on the properties to be exchange. And by experience I can assure you that you do not want an automatic selection if the grids do not match.
    - \* Response: There is no automatic selection of the regridding method, but the setup depending on the grid types is done automatically. We will make this more clear.
- using Python
  - this is not a unique feature
    - \* Response: True, but again, we didn't want to list only unique features but state why the combination of features provided by FINAM makes it unique in its whole. See below.

**Response:**

We think there was a misunderstanding with this paragraph, since we didn't want to state that all the mentioned features are unique on their own but that the selection of features provided by FINAM makes it unique in its whole. We will clarify this in the revised version of the manuscript.

**Manuscript changes:**

We added the following at the end of the paragraph: "These features may not be unique in isolation, but in their combination, making FINAM a flexible and easy to use solution for the coupling of environmental and other models."

Regarding regridding, we added the following to section 3.3: "The regridding method to use needs to be specified by the user and is depending on the exchanged data." Figure 1:

Original comment: repetition of ", then" Response: We will fix the repetition of ", then". Manuscript changes: Replaced the second "then" by "next".

**L95:**

**Original comment:**

repetition of "handled by" maybe: "number of in- and outputs handled by in- and output slots respectively"

**Response:**

We will update this line to improve clarity and avoid repetition.

**Manuscript changes:**

Changed to "an arbitrary number of inputs and outputs, handled by input slots and output slots, respectively"

**L96:**

**Original comment:**

*"but do not have to" maybe: "an optional internal time step"* **Response:**

We will rephrase this for better clarity, using the suggested "an optional internal time step."

Manuscript changes:

Rephrased as suggested.

**L99:**

**Original comment:**

"when receiving input" maybe: "when input is available" **Response:**

We will rephrase this for improved clarity.

Manuscript changes:

Rephrased to "... they are executed when new input becomes available, or when an output is requested, respectively."

**L128:**

**Original comment:**

"Data exchange [...] takes place purely in memory" How is data exchange in case no adapter is used (by value or reference)? How do you avoid data copies? Is this a concern at all?

**Response:**

Due to the premise that models are assumed to be developed independently we can not require these models to use the same data representation internally and thus need to copy data when pushed for one model to another. In general we avoid copying of data internally if possible. Also, since we do not aim at HPC explicitly this is not really a concern at the moment.

Manuscript changes:

None.

**L122:**

**Original comment:**

"pint quantities" This is not a common term for me. Maybe something like: "unit handling is automatically done by the pint library"?

**Response:**

We will clarify the use of the "pint" library for unit handling.

**Manuscript changes:**

Rephrased to "wrapped in quantities provided by the  $pint^2$  library, which handles units automatically."

**1.3.3 Section 2.1.4:**

**Original comment:**

The scheduling algorithm is only required because FINAM runs on a single process and components can have varying time steps. If all components were to run on dedicated processes, this should be much simpler. Or not? You could add this information here to explain why other coupling solutions do not have such a complex scheduling algorithm.

**Response:**

As you already noted, we allow models to have a flexible time-step which was a prerequisite for FINAM. We do not expect, that a parallel implementation would simplify the mediation between models running on different flexible timesteps, although it is true that running components on dedicated processes with getters and setters in the code removes the need for explicit scheduling when time stepping is fixed and compatible. Also, this scheduling has the advantage that in circular couplings it is explicitly defined which model uses data from a past time-step, that would happen implicit otherwise.

**Manuscript changes:**

We added the following paragraph to the section: "The scheduling algorithm is mainly required for two reasons: firstly, it allows for the coupling of models

2. https://pint.readthedocs.io

with arbitrary, potentially incompatible time steps, and even for model time steps varying over the course of a simulation run. Secondly, it allows models to use input for the upcoming time step, instead of the past one."

**1.3.4 Section 2.1.5:**

**Original comment:**

Again this is only required due to the serial nature of FINAM. Maybe add a small hint on why this approach is needed as compared to a parallel implementation.

**Response:**

This has nothing to do with scheduling or parallelism and is more a unique feature of FINAM. Iterative initialization comes from FINAMs ability to exchange meta-data and data specifications in any direction (up- and downstream). If a model needs meta information from a connected model, one could of course define these meta information twice in both involved components but this would make it impossible to have adaptively set up components (e.g. a data generator can get the required grid automatically from the data consumer). Also, this initialization can be different for each in- and output of one model with complex dependencies on other components. In summary, this mechanism protects the user from specifying already available metadata over and over again. We we will update this section slightly to highlight that this is a unique feature of FINAM and add a simple example.

**Manuscript changes:**

No changes, as we already explain why FINAM uses the iterative initialization algorithm in: "All of these examples require the exchange of metadata between components (and adapters), potentially in both directions. To make this possible in an automated way and without requiring a user to manually set all metadata, FINAM uses an iterative initialization process"

Figure 7:

Original comment: "update and the" Response: We will rephrase this. Manuscript changes: Fixed to "updated".

**1.3.5 Section 2.2:**

**Original comment:**

For a non-Python model the implementation of a FINAM wrapper might not be trivial. Maybe you could give more information on that (e.g., add getter

**routines for grid information, add Python bindings for C or Fortran routines using Cython, ...). The FINAM wrapper for mHM could serve as an example. **Response:**

We will elaborate on the process of wrapping non-Python models for FINAM and provide more detailed examples, such as mHM (https://mhm-ufz.org), to give the reader a better understanding of the effort involved. But we didn't want to be too technical in this paper. More detailed explanations are given in the FINAM book for the interested reader. But it is true, that a better overview of this process could enhance the manuscript.

**Manuscript changes:**

We added the following paragraphs to the section:

"Depending on the code structure of the model, some refactoring may be required to provide separate routines for initialization and model stepping."

"Python bindings can be created using libraries such as Cython (Behnel et al. 2011), scikit-build (Fillion-Robin et al. 2018), f2py (Harris et al. 2020), pybind11 (Jakob et al. 2017),  $pyo3^3$ ,  $ctypes^4$ ,  $swig^5$ , or  $cffi^6$ ."

**L181-184:**

**Original comment:**

Using the line count for the trivial PET example may be misleading. Again mHM could be a more realistic example.

**Response:**

We still think that, to illustrate the structure of a FINAM component, the PET example is very well suited. The mHM component doesn't look much different, it is just more bloated due to the much greater amount of inputs and outputs. The key point is, that the mHM component uses the mHM wrapper that is provided in the mHM code base and that both components, PET and mHM, just call a function in their update method to calculate the states for the current time step. We will add some clarification about what the thin line is between a FINAM component wrapper and the Python wrapper of the native model.

**Manuscript changes:**

We extended the section by a paragraph discussing the creation of the Python bindings of mHM to highlight the difference between the bindings and the FINAM wrapper: "As an example, we want to discuss the Python bindings for the mesoscale hydrological model - mHM (Samaniego et al. 2010; Kumar et al. 2013) written in Fortran. The developers had to take these steps: (i) encapsulating the time loop in a callable subroutine by copying over code, (ii) encapsulating the initialization and finalization of the model by separating the main driver of mHM into callable subroutines, and (iii) writing an f2py wrapper that links against the mHM library and provides routines to call the mentioned initialization, update, and finalization subroutines as well as routines to retrieve and alter the internal states. This straightforward refactoring was done

3. https://github.com/PyO3/pyo3

4. https://docs.python.org/3/library/ctypes

5. https://www.swig.org

 $6.\ https://github.com/python-cffi/cffi$

in a manageable amount of commits and a positive side effect is that mHM is now installable via  $pip^7$ ."

**L194:**

**Original comment:**

VTK, and CSV These are common formats (in your community)? **Response:**

Yes. OpenGeoSys (OGS) for example produces VTK files (among others) and Formind for example produces CSV files. CSV is a common table file format that suits well for non-spatial data input and output.

Manuscript changes:

None.

**L194:**

**Original comment:**

real-time visualization Do you support online visualization with Catalyst/Paraview? **Response:**

Not at the moment. The live plotting features are provided as components in the finam-plot module and are based on matplotlib.

**Manuscript changes:**

Amended by mentioning the visualization package: "Live plotting capabilities provided by finam-plot and finam-graph, which are based on matplotlib, enable real-time visualization..."

**1.3.6Section 3.1:**

**Original comment:**

Depending on your targeted user group bidirectional exchanges between models may be a regular coupling use-case and not worth writing a dedicated section about (see "OASIS3-MCT User Guide" section "2.1 Configurations of components supported").

**Response:**

Since bi-directional coupling is a basic use-case for coupling tools, we think this is very well suited to be featured in an example. We again want to highlight that the important part here is the use of the delay adapter to solve the circular dependency introduced by both models.

Manuscript changes: None.

7. https://git.ufz.de/mhm/mhm/-/tree/v5.13.1/pybind

**L245:**

**Original comment:**

"This ease of use contrasts sharply with traditional frameworks, which often require significant effort to configure and manage bidirectional interactions." This again depends on your perspective and usage of a coupling solution. The provided example could be probably written in a similar fashion using the latest YAC version.

**Response:**

From our perspective ease of use is not only measured in written lines of code but also in the provided documentation and user guidance which equip users with confidence to tinker around with model configurations.

**Manuscript changes:**

The section was reworked. The sentence "This ease of use contrasts sharply with traditional frameworks, which often require significant effort to configure and manage bidirectional interactions." is not necessary for the understanding and led to misunderstanding. Hence it is deleted. Instead, the section now focuses more on the fact that bidirectional coupling is easy to implement.

**1.3.7 Section 3.3:**

**Original comment:**

Do you make a difference between grids defined in Cartesian space and on the sphere? Are they compatible?

**Response:**

The coordinate reference system (CRS) is part of the Grid definition in FINAM to account for Cartesian and spherical coordinates. Coordinate transforms are performed automatically if grids have different CRS. We will add a short note to clarify this in the manuscript.

**Manuscript changes:**

Added note to the first paragraph in section 3.3.: "In FINAM the coordinate reference system (CRS) is part of the grid definition to account for Cartesian and spherical coordinates. Coordinate transforms are performed automatically when grids have different CRSs."

**L300-303:**

**Original comment:**

Regridding methods do not only depend on the grid specifications, but also on the type of fields to be exchanged. The correct regridding method and its configuration (e.g., for conservative interpolation: order and normalization) can have a significant impact on the coupling results. While using default configuration can get plausible results quickly, more detailed configurations can often lead to better results.

**Response:**

That is of course true. What we meant is, that users don't need to know the technical aspects of regridding. The method to use still needs to be explicitly set

by the modeler. We will expand on the considerations for choosing regridding methods based on grid specifications and field properties.

**Manuscript changes:**

The corresponding section was reformulated. It is pointed out that the transformation between different CRS is handled automatically: "Performing as a dynamic adapter, it automatically detects the specifications of the source and target grids and derives the required transformation for differing CRSs. The regridding method to use needs to be specified by the user and is depending on the exchanged data. Despite the underlying complexity of this task, we have engineered the tool to be user-friendly, ensuring that its advanced capabilities are accessible without the need for detailed technical knowledge of regridding."

**L316-320, Figure 19, and Figure B1:**

**Original comment:**

This could give the impression that FINAM introduces conservative interpolation, while it actually is using a basic functionality provided by ESMF. This could be made clearer in my opinion. I do not see the added benefit to the paper of Figure B1.

**Response:**

In line 295 we already state that finam-regrid wraps the functionalities of ESMF but we will mention again in line 318 that the conservative regridding really comes from ESMF. We will ensure that Figure B1's role is clear. Even though we use already existing solutions to the problem, we still think that this important part of a coupling framework should be showcased. Therefore we would keep the plot to round off the presentation of FINAM.

**Manuscript changes:**

We amended L318 by clarifying that the conservative regridding uses ESMF.

**1.3.8 Discussion:**

**Original comment:**

As mentioned before, without a clearer definition of the targeted users and use-case, different conclusions could be drawn. A couple of the points made in this section are not unique to FINAM, but are characteristics of coupling frameworks in general. It could improve the discussion if the paper makes it clearer where FINAM takes advantage of general coupling framework concepts and where it deviates from other implementation in order to generate different traits.

**Response:**

As mentioned before, we will clearly define the target users and use cases of FINAM to avoid generalizations that may apply to other frameworks as well.

**Manuscript changes:**

We clarified FINAMs target users and use cases in the introduction. See above.

**L328:**

**Original comment:**

"Python as the foundational language" From my point of view, this only makes sense if the paper defines clearer goals. Your description would also describe some of the requirements for coupling global climate models, but in that case additional requirements on compute performance, memory handling and memory consumption might lead to a different choice (e.g., C for YAC).

**Response:**

With the update of the target group and the clearly defined aim for rapid prototyping the choice of Python will be more comprehensible.

**Manuscript changes:**

We clarified FINAMs target users and use cases in the introduction. See above.

**L336-337:**

**Original comment:**

"The coupler makes use of this characteristic by requiring that each model needs to be able to perform a single time iteration" Maybe: "The coupler makes use of this characteristic by requiring that the user exposes the execution of the individual time iterations through an interface routine."

**Response:**

We will modify this to describe how users expose time iterations through interface routines.

**Manuscript changes:**

None. Internal evaluation showed that the current wording is more comprehensible for a wide audience.

**L341-345:**

**Original comment:**

Do any of the coupling solutions to which FINAM is compared to use a pure pull-based approach or where does the comparison to this approach come from? Was this a potential option for implementing FINAM?

**Response:**

From our point of view, YAC for example is a pull-based solution (although with YAC, components simply run and are not only executed when requested for data), which is also why it does not require an explicit scheduling algorithm. We already explained in detail (L342-345), that we wanted features that can't be achieved with a pull based approach. We will amend the paragraph by an explanation why push based components are valuable, like flexibly timed data-writers.

**Manuscript changes:**

We added the following to the end of the section: "Finally, the hybrid approach allows for push-based components that react to new data becoming available."

**L346:**

**Original comment:**

"traditional coupling methods" Maybe: "coupling libraries approach-based solutions like OASIS"

**Response:**

We will rephrase this for better clarity.

Manuscript changes:

We rephrased to: "In contrast, solutions based on coupling libraries like OASIS often involve the integration of routines in their supported languages directly into the model source code, ..."

**L348:**

**Original comment:**

"must ensure data compatibility with respect to units" In my experience that has never been an issue.

**Response:**

We will revise this statement to reflect that unit compatibility might not be an issue for all coupling libraries. But from our experience this was a huge source of problems in earlier works.

**Manuscript changes:**

We weakened our statement by replacing "this method imposes" by "this method may impose".

**L348:**

**Original comment:**

"must ensure data compatibility with respect to  $[\ldots]$  grid definitions" For me that is a basic task for a coupler to make sure, that the user does not have to take care of this. In the case of YAC, all grid combinations are compatible.

**Response:**

We will clarify that grid compatibility checks are meant to indicate the need for a regridding adapter, which is not required when grids are equal or when the transformation is trivial (like transposition or axis flipping). Examples for compatible grids are: (i) from NetCDF: grids that have reversed axes order (zy-x) and a potential "bottom-up" axis (y axis with decreasing values), and (ii) from VTK grids: grids that always use x-y-z order with increasing axes. These grids can define the same spatial layout but the data still requires (trivial) transformations, like transposing and flipping, when mapped from one grid to the other.

**Manuscript changes:**

We amended the subsequent paragraph to clarify the meaning of grid compatibility: "Regridding is not required if grids are compatible, in the sense that only trivial transformations like axis flipping or transposing are needed."

**L348:**

**Original comment:**

"must ensure data compatibility with respect to (...) time reference" In my experience that has never been an issue; at least for the application YAC has been used for till now. This may be different for your targeted use-case.

**Response:**

We will clarify that time reference issues may vary depending on the application and use case. The problem here are different variable categories like intensive (time-point related) and extensive (time-span related) variables (terms following the CF-Conventions). The point is that FINAM utilizes a unified meaning to state what a time-stamp refers to.

**Manuscript changes:**

None. We already explain this in detail starting in L385.

**L349:**

**Original comment:**

"necessitating extensive modifications to the original model code" Yes, coupling code is directly included in the model code, but with a good software design it can be encapsulated, and the resulting impact can be minimal. On the other hand, rewriting an existing model into a form that is usable by a coupling framework can be a huge amount of work, which you also mention later in this section. I would not categorize any of these approaches as superior over the other or requiring more or less work than the other. It primarily depends on the specific application. You are in the particular position of having a model (mHM) for which both approaches have already been implemented. Maybe you could compare the two implementations in terms of measurements like lines of code, man-hours, complexity, or ease of maintenance.

**Response:**

We will acknowledge that the effort to restructure models for coupling frameworks varies by application and potentially mention examples from our previous work. What we want to mention here, as already written above, is that modifications to the code-base don't require FINAM specific types or methods that would require linking your code against an external library like it would be the case with ESMF, OASIS or YAC. We basically need a routine to initialize the model and one routine to do one internal time-step. Then you can write a thin python wrapper that is able to call these functions and provides a mechanism to retrieve and set internal states between the time-step calls. With mHM, the experience was that the refactoring to have the time-loop call in an encapsulated subroutine that advances the model one time-step was beneficial for the whole code-base (and not too demanding). Whereas the incorporation of YAC, that came after the refactoring of mHM for FINAM, benefited from the then better structure. But with YAC we had to write a separate driver to include the YAC specific calls. From this point it is hard to compare the effort required to incorporate YAC and to make mHM FINAM ready, but the code-base benefited more from the general refactoring than from incorporating YAC specific routines (that also introduced more conditional compilation). The same statement is true for other models that were made ready for FINAM (like OGS, Formind or Bodium) since all improved their internal structure. Generally speaking, if a model already has a good code structure (initialization and stepping in subroutines), it is almost already ready for FINAM. Everything else that is then required are Python-bindings as described in the manuscript and these do not infer with the source code (e.g. using tools like scikit-build (https://scikitbuild.readthedocs.io) to create Python packages with cmake based extensions).

**Manuscript changes:**

We weakened our statement by using "potentially necessitating extensive modifications" instead of "often necessitating extensive modifications".

**L351-352:**

**Original comment:**

"minimizing the need for direct alterations to model code" This is only true after the initial restructuring for a coupling framework has been done.

**Response:**

This is of course true. We will clarify that minimizing direct alterations to model code is a benefit after the initial restructuring as described in the response above.

**Manuscript changes:**

We added the following to the end of the paragraph, referring to the extended section "Wrapping models": "However, as described in Section 2.2, some refactoring might be necessary."

**L354-355:**

**Original comment:**

"regridding adapters automatically determine their required transformation from the connected source and target components" As mentioned before, from my experience this should not be done automatically, because the correct configuration of the regridding depends on a variety of factors (e.g., grid types, properties to be exchanged, masks,...).

**Response:**

We mean that users don't need to explicitly specify source and target grid of the adapter, but that the adapter obtains those from the linked components during the iterative connection phase described above. Details of the regridding, like the used method, are still configured by the user. We will clarify this in the text.

**Manuscript changes:**

In the section on regridding, we clarified what is specified by the user and what is automated. See the respective comments.

**L361-364:**

**Original comment:**

"parallelization" This is potentially a very important topic and should get more explanation on what is planned for FINAM (e.g., support for multiple serial components running on different processes, supporting MPI-parallelized components, supporting compute and memory intensive highly parallel components). I think the authors underestimate the implications of supporting parallelism in coupled setups. To give an example: handling an ICON R02B11 grid is a challenge in itself. It contains more than 300 million cells. Having a whole copy of the grid available on a single process is often impossible. This makes online computation of the regridding weights in a reasonable amount of time very difficult.

**Response:**

With the better specification of the target user group, as mentioned several times above, this will be seen in another light since highly parallelized applications are currently not the target for FINAM and we are glad that YAC is there to help. We agree that supporting parallelism poses significant challenges. We identified parallelization as an important issue that we need to tackle in the near future.

**Manuscript changes:**

We are now more explicit on FINAMs target use cases. See comments on the introduction.

**L364-365:**

**Original comment:**

"Additionally, making a model ready for FINAM can require a significant investment of time and resources." As mentioned before, this could somewhat be quantified using mHM as an example.

**Response:**

As stated above, it depends on the state of the code-base of the model. If the code is well structured and encapsulated, the effort can be minimal. This is for example the case if your model provides a basic model interface (BMI) discussed above. We will clarify what is required for non-Python models (see mHM details above). The remaining task is to write a Python wrapper, where an abundance of tools exist to realize this as mentioned in the manuscript.

**Manuscript changes:**

We extended section "Wrapping models" to clarify the potentially required refactoring, as well as the separate tasks of providing Python bindings and writing the wrapper.

**L383:**

**Original comment:**

"A unique feature of FINAM is its support for bidirectional coupling by temporal delaying circular input/output connections" The unique feature of FINAM is the ability to do this explicitly (I do not know whether ESMF supports this). However, this can be done implicitly with a coupling library.

**Response:**

We will clarify that while FINAM supports explicit bidirectional coupling, some coupling libraries may achieve this implicitly.

**Manuscript changes:**

We extended the sentence: "While other couplers do this implicitly based on the code position of their getters and setters, FINAM provides adapters to explicitly control the time offset of the exchanged data."

**L387:**

**Original comment:**

"FINAM standardizes time representation across models" All coupling solutions have a standardized method on how to define time. The way FINAM makes use of it is unique (at least compared to OASIS and YAC; I do not know how it compares to the other coupling solutions mentioned above).

**Response:**

We agree that this is not a unique feature for a coupling library and we never state that FINAM is special there. But we still need to mention the way it is done somewhere in manuscript and therefore we would like to keep this as it is.

Manuscript changes: None.

**L407:**

**Original comment:**

"forward-thinking approach" This implies a backwards-thinking approach for other coupling solutions... Maybe: "an approach better suited for ..."

**Response:**

We will rephrase "forward-thinking approach" to avoid implying that other solutions are outdated.

**Manuscript changes:**

We rephrased the statement: "In summary, FINAM provides a flexible and user-oriented approach to model coupling, leveraging Python's versatility to integrate a wide range of models."

**L409-410:**

**Original comment:**

"FINAM represents a significant advancement in the field of computational modeling." I do not think that FINAM is practical for all coupling use-cases. Therefore, I would not support this general statement.

**Response:**

With an improved target user specification it will be more clear that FINAM is not the solution to all problems related to coupling. In its field of application we still think it is a significant advancement with providing rapid prototyping and a huge set of well documented tools.

**Manuscript changes:**

See updates on the Introduction above.

**Chapter 2**

**Reviewer: Jannes Breier**

**2.1 General Comments:**

**Original comment:**

This manuscript was forwarded to me by a colleague of mine who knew that I am working on a similar but more applied model coupling library for a large-scale biosphere model using a different coupling approach. Based on this experience and the knowledge that many modelers want to couple different models without having to develop the corresponding software themselves, this endeavour, especially in its generalised form, is a highly significant scientific contribution. In particular, the structure, underlying grammar, and logic of the coupling framework and the way it is built up in the concept of components speak for themselves. Some of the results are easily reproducible and the framework is documented in an exemplary manner, especially through the linked API documentation in the Code Availability chapter.

**Response:**

Thank you for your kind words and support. We are glad you find the structure and documentation of FINAM exemplary.

**2.2 Specific Comments:**

**Original comment:**

What is the purpose of overloading the bitwise operator  $(\dot{z}\dot{z})$  in the code? This is a very unique feature and not clearly explained in the text.

**Response:**

This is basically syntactic sugar to make it visually more pleasing to write these scripts. This is already indicated in the caption of figure 5 ('Data connections are denoted by the overloaded bit shift operator " $i_{i}i$ " (for visual reasons).'). We will explain in more detail that overloading the bitwise operator ( $i_{i}i$ ) allows for more concise coupling code, compared to nested method calls that would be required otherwise.

**Manuscript changes:**

We added a short paragraph to the "Linking components" section: "For visual reasons, we overrode the bit shift operator ">>" to create links between an output of one component and an input of another component. This makes the coupling configuration more readable compared to chained calls of linking methods."

**Original comment:**

What would be an example for two models written in different languages, say one in Python, the other one in C++? How would the coupling work in this case? This is a feature which is mentioned in the text but not explained in detail.

**Response:**

We will expand on the coupling between models in different languages. We will clarify that from the perspective of FINAM, all models are just components written in Python. Models in other languages realize these components by the use of Python bindings for the respective model. Due to the use of Pythonbindings for each model, the underlying language in use is irrelevant and there is a huge set of tools to establish these bindings as mentioned in the manuscript.

**Manuscript changes:**

Based on the comments from the other reviewer, we expanded the section "Wrapping models" to give a more detailed example for a wrapped model in another language (mHM written in Fortran).

**Original comment:**

What are the limitations of the regrid functionality, how are things handled like different geographical extents, different geographical coordinate systems, projections, different longitude latitude resolutions, etc.?

**Response:**

We will expand on the use of ESMF for our regridding adapters and make clear that the functionality and limitations of ESMF apply here. Coordinate transformations between different CRS are done automatically. We will highlight this in the revised manuscript.

**Manuscript changes:**

Based on the comments from the other reviewer, we expanded the section regridding example and made it more clear that all the functionality from the ESMF regridders is just forwarded to **finam-regrid** and that CRS related transformations are done automatically.

**2.3 Final Remarks:**

**Original comment:**

It is noteworthy that the important limitations of the framework, such as the limitations in the parallelisation of models via MPI, are discussed in the text. Even though this might limit the applicability of the framework in some cases, it does provide a clear and well documented path for others that want to couple models for novel research questions. Therefore, this manuscript is highly recommended for publication in GMD.

**Response:**

We appreciate your acknowledgment of FINAM's limitations, especially regarding parallelization, and are glad you still see it as a valuable contribution.

**Bibliography**

- Bauer, T. P., P. Holtermann, B. Heinold, H. Radtke, O. Knoth, and K. Klingbeil. 2021. "ICONGETM v1.0 flexible NUOPC-driven two-way coupling via ESMF exchange grids between the unstructured-grid atmosphere model ICON and the structured-grid coastal ocean model GETM" [in English]. Publisher: Copernicus GmbH, Geoscientific Model Development 14 (8): 4843–4863. Accessed December 16, 2024. https://doi.org/10.5194/gmd-14-4843-2021.
- Behnel, S., R. Bradshaw, C. Citro, L. Dalcin, D. S. Seljebotn, and K. Smith. 2011. "Cython: The Best of Both Worlds." Conference Name: Computing in Science Engineering, *Computing in Science Engineering* 13 (2): 31–39. https://doi.org/10.1109/MCSE.2010.118.
- Buis, S., A. Piacentini, and D. Déclat. 2006. "PALM: a computational framework for assembling high-performance computing applications" [in en]. \_Eprint: https://onlinelibrary.wiley.com/doi/pdf/10.1002/cpe.914, Concurrency and Computation: Practice and Experience 18 (2): 231–245. Accessed December 12, 2024. https://doi.org/10.1002/cpe.914.
- Chourdakis, G., K. Davis, B. Rodenberg, M. Schulte, F. Simonis, B. Uekermann, G. Abrams, et al. 2022. "preCICE v2: A sustainable and user-friendly coupling library [version 2; peer review: 2 approved]." Open Research Europe 2 (51). https://doi.org/10.12688/openreseurope.14445.2.
- Fillion-Robin, J.-C., M. McCormick, O. Padron, M. Smolens, M. Grauer, and M. Sarahan. 2018. jcfr/scipy\_2018\_scikit-build\_talk: SciPy 2018 Talk scikit-build: A Build System Generator for CPython C/C++/Fortran/Cython Extensions. Accessed July 20, 2024. https://doi.org/10.5281/zenodo.2565368.
- Harris, C. R., K. J. Millman, S. J. van der Walt, R. Gommers, P. Virtanen, D. Cournapeau, E. Wieser, et al. 2020. "Array programming with NumPy." *Nature* 585:357–362. https://doi.org/10.1038/s41586-020-2649-2.
- Hutton, E. W., M. D. Piper, and G. E. Tucker. 2020. "The Basic Model Interface 2.0: A standard interface for coupling numerical models in the geosciences." Publisher: The Open Journal, *Journal of Open Source Software* 5 (51): 2317. https://doi.org/10.21105/joss.02317.
- Jakob, W., J. Rhinelander, and D. Moldovan. 2017. pybind11 Seamless operability between C++11 and Python.

- Kumar, R., L. Samaniego, and S. Attinger. 2013. "Implications of distributed hydrologic model parameterization on water fluxes at multiple scales and locations" [in en]. Water Resources Research 49 (1): 360–379. Accessed March 9, 2020. https://doi.org/10.1029/2012WR012195.
- Samaniego, L., R. Kumar, and S. Attinger. 2010. "Multiscale parameter regionalization of a grid-based hydrologic model at the mesoscale" [in en]. Water Resources Research 46 (5). Accessed March 9, 2020. https://doi.org/10. 1029/2008WR007327.

---

## Author Response (AR2)

**2. Rebuttal**

The second point-by-point response to the reviews including a list of all relevant changes made in the manuscript *"FINAM - is not a model (v1.0): a new Python-based model coupling framework"*.

Sebastian Müller        Martin Lange        Thomas Fischer

Sara König        Matthias Kelbling        Jeisson Javier Leal Rojas

Stephan Thober

February 25, 2025

**Chapter 1**

**Reviewer: Anonymous referee #3**

**1.1 General Comments:**

**Original comment:**

*It is an honor to review such a technically oriented research paper. All the papers I have reviewed before focused on scientific issues, and technical problems were usually solved by researchers themselves while addressing scientific questions. However, this paper attempts to solve many common problems that researchers often encounter and provides an interesting solution. I am not professionally engaged in developing coding tools, so I may not fully understand some aspects, but I believe this work is meaningful and can provide convenience for researchers.*

**Response:**

Thank you for your positive feedback and for recognizing the value of our approach. Our main goal is in fact to simplify and automate aspects of model coupling that are often cumbersome in research workflows. We appreciate that you find this work meaningful and convenient for researchers and hope that our explanations and examples clarify the technical details for a broader audience.
* * *
**1.2 Specific Comments:**

**Original comment:**

*1. In my own research, I often deal with model coupling, such as using the output of one model as the input for subsequent models. However, many models are encapsulated, and we cannot access their source code. Such models should not be couplable in the FINAM system, which limits the application scope of FINAM.*

**Response:**

We acknowledge that fully encapsulated models can be challenging to couple if no mechanisms exist to expose their inputs and outputs. FINAM's approach

relies on 'wrapping' models with a Python-based interface, where at least a portion of the model's functionality (e.g. initialization, stepping, data exchange) is accessible. If the model is entirely closed source or offers no means of extracting or injecting data programmatically, it cannot be directly coupled within FINAM. However, as some users employ scripting interfaces or configuration-based I/O even for closed source applications (e.g., controlling runs via command-line arguments, batch scripts or APIs), these can sometimes be wrapped with a 'blackbox' style component in FINAM. In other words, if there is any programmatic handle - although minimal - we can integrate that into FINAM with suitable wrappers. We already mentioned file-based couplings as the first approach to combine models in the Introduction.

**Manuscript changes:**

We added a clarifying paragraph (Section 2.2): "If no Python bindings of the model exist, but it can be run as a black box for a single time-step, there is also the possibility to create a component that prepares the required input files for each time-step, calls the model, and reads the output files to provide the data in the FINAM composition. But be aware that this approach may introduce performance bottlenecks since it is basically a file based coupling."
* * *
**Original comment:**

*2. Through the example in Figure 1, I indeed grasped the purpose of the FINAM tool. In similar computations, it is often necessary to read data, output intermediate variables, read them again for further calculation, and then output the results. This is a very cumbersome process. The FINAM tool has been encapsulated into a Python package, eliminating a lot of intermediate work and directly outputting results. I wonder if this process can also be edited using Python's parallel computing syntax to achieve multithreaded computation?*

**Response:**

FINAM is currently designed for serial execution and does not natively support parallelization through multithreading or MPI as already mentioned. However, one could think about integrating Python-based parallel computing approaches (e.g., using `multiprocessing`, `joblib`, `dask`, etc.) around FINAM if certain components are naturally parallelizable. In that scenario, FINAM would still manage the data flow between components, while parallel execution of selected tasks could be handled at the Python level or by the native models themselves. We note that true multithreaded speedups in Python can be limited by the Global Interpreter Lock (GIL), unless one uses libraries that release the GIL (e.g., NumPy with native code sections). Full MPI-based parallelization is currently out of scope, but we plan to investigate how FINAM can interface with parallel libraries in future work as mentioned in the discussion.

**Manuscript changes:**

We added the following paragraph in the Discussion: "There are also Python-based parallelization approaches (multiprocessing[1], joblib[2], dask[3], etc.) that could be used in the future to run independent parts of the composition in parallel."
* * *
1. https://docs.python.org/3/library/multiprocessing.html
2. https://joblib.readthedocs.io
3. https://www.dask.org/
* * *
**Original comment:**

*3. Although it is difficult to directly couple encapsulated models with other models, it seems possible to use syntax to drive the model for computation, output results, and then couple the result file with other models. If this can be operated within the FINAM framework, it could also reduce simulation time to some extent.*

**Response:**

Yes, a "file-based" or "black-box" approach can be used for models that do not expose sufficient source code or direct I/O routines. In this scenario, a FINAM component wrapper orchestrates the external model run by generating the necessary input files, executing the model, and reading its output files back into FINAM for subsequent coupling steps. While this still involves intermediate file I/O, it can be streamlined by a single controlling workflow in Python, potentially saving user time and reducing the complexity of manual data handling. However, the simulation time benefit depends heavily on how frequently data needs to be exchanged. For very frequent exchanges, file-based I/O may become a bottleneck. FINAM aims to be flexible enough to accommodate this style of coupling, but its primary design still favors in-memory data transfer for accessible source code or APIs.

**Manuscript changes:**

We expanded Section 2.2 as described above.
* * *
**Original comment:**

*4. In the field of hydrology, surrogate or alternative models have been very popular recently. A complex model can be learned using regression algorithms to understand its inputs and outputs, and then the constructed regression model can be used as an alternative model. It appears that coupling alternative models directly with FINAM would be much easier.*

**Response:**

We agree that surrogate modeling is becoming increasingly prominent, particularly in hydrology and other domains where computationally heavy models can be approximated by faster regressions or machine learning algorithms. FINAM's Python-centric design is well-suited to integrating such surrogate models, as these models often come in the form of Python packages or can be accessed via scikit-learn, TensorFlow, PyTorch, etc. Wrapping a surrogate model in FINAM typically involves creating a wrapper that runs the regression or neural network for each time step, and the rest of FINAM handles data exchange with other components. This can be done in the exact same way as we described the PET component. We see this as a major advantage for users who wish to experiment with hybrid or alternative modeling approaches without altering the underlying code of the original more complex model. Since the model structure is not relevant for FINAM we do not want to add another paragraph for specific types of models, since any time-step-based model is suitable to be coupled with FINAM.

**Manuscript changes:**

In the Summary, we now acknowledge that surrogate models are candidates for couplings.
* * *
**Original comment:**

*5. It is also currently popular to couple machine learning models with traditional models. Could machine learning be considered for future development?*

**Response:**

Yes, machine learning (ML) integration is one possible use case for FINAM. Since FINAM is built on Python, it is relatively straightforward to couple ML models and traditional numerical models within the same workflow since they only need to be wrapped in a component. As the internal model structure (like process-based or ML-based) is irrelevant for FINAM, we would like to not bloat the text further with specific sorts of models or give the impression FINAM has a special relation to machine learning.

**Manuscript changes:**

In the Summary we now acknowledge that ML models are candidates for couplings.
* * *
**Chapter 2**

**Reviewer: Nils-Arne Dreier**

**2.1 General Comments:**

**Original comment:**

*The article "FINAM - is not a model (v1.0): a new Python-based model coupling framework" describes a novel coupling framework whose main focus is to simplify the coupling configuration, with Python as the fundamental language for composing model components. The authors provide a detailed explanation of the concepts of FINAM and explain the decisions made during the development process. Furthermore, they provide examples of how to use FINAM.*

*This article is fitting within the journal's context and is of high quality, hence, my recommendation for its publication with minor revisions.*

**Response:**

Thank you for your positive feedback and your recommendation for publication. We are pleased that you find our approach and detailed explanations aligned with the scope of the journal.
* * *
**2.2 Specific Comments:**

**2.2.1 Introduction:**

**L5:**

**Original comment:**

*"such as YAC, ESMF, or OASIS" I wondered if there was a specific reason behind the order of the stated "coupling solutions". If not, it might be advisable to list them alphabetically. The same applies to their mentions later in the text.*

**Response:**

Thank you for pointing this out. There is no particular reason for the existing order. We will adjust the text to list them alphabetically for consistency and clarity. Where these coupling solutions are cited again, we will maintain the alphabetical order.

**Manuscript changes:**

In the Abstract and other relevant sections, we changed the list to: "ESMF, OASIS or YAC." In other places like the Introduction we kept the order (OASIS, YAC and ESMF) since this is the logical order to introduce FINAM (first plain couplers, then frameworks).
* * *
**2.2.2 Section 2:**

**Original comment:**

*By the conclusion of this section, I became curious about any other adapters and utilities that FINAM provides to formulate coupled experiments. While I found this information in the online documentation, it would be helpful if the authors could include a brief summary of the key adapters and utilities towards the end of this section to offer a deeper understanding of the potent adapter concept.*

**Response:**

We appreciate this suggestion. We realize that providing a short overview of existing adapters and utilities within the manuscript clarifies the potential of FINAM for new readers without requiring them to consult external documentation. We will briefly summarize the available adapters (e.g., for file-based I/O, time shift/delay, regridding) and mention how new adapters can be implemented.

**Manuscript changes:**

We added a subsection "2.3 Key Adapters and Utilities" that lists commonly used adapters (e.g., DelayFixed, Regrid, FileIO) and briefly describes their functionalities, referencing the online documentation for more detailed usage examples.
* * *
**L151:**

**Original comment:**

*"When units are not equivalent, like L/m2 and mm, but compatible, like K and °C, they are converted automatically." I'm a bit confused with the units here:*

- *I don't understand why $L/m^2$ (volume/area = length) and mm (length) are not equivalent?*

- *The authors might need to explicitly define what they mean by "equivalent" and "compatible".*

**Response:**

Thank you for highlighting this confusion. The original wording was indeed ambiguous. In our implementation, we differentiate between units that require a conversion (e.g., °C to K), and those that are essentially the same (e.g., $L/m^2$ and mm).

**Manuscript changes:**

We updated the sentence to: "Data with compatible units such as K and °C will be automatically converted. Equivalent units such as $L/m^2$ and mm will not cause a conversion."

**2.2.3   Section 3.1:**

**Original comment:**

*From my understanding, integrating the "DelayFixed" adapter changes equation (2) so that $sm(t-1)$ instead of $sm(t)$ is used. This should be explicitly highlighted.*

*I was also wondering whether the system of equations could be solved without modifying the actual timestepping scheme, e.g. by using a fixed-point iteration or Newton solver. This would require recomputing timesteps of a component or compute differentials. I understand that this goes beyond the scope of this paper and could be considered in future work.*

**Response:**

We agree that the "DelayFixed" adapter effectively shifts the time dependency so that the components use the state of the previous time step instead of the current one. This is essential for modeling lagged dependencies or simulating bidirectional couplings where circular references would otherwise appear.

Regarding the possibility of solving the system via fixed-point iterations, Newton solvers, or more advanced numerical schemes, we think this would imply a huge impact on model developers since models would need to be able to save their state and reset if needed. As our focus is on models that were developed for stand-alone usage, we think this to be a major requirement that would raise the bar to high for many users. FINAM currently focuses on explicit time stepping at the component level, but an extension to iterative or implicit coupling methods might be valuable for problems requiring tighter convergence or strongly coupled feedbacks but would need an investigation how to incorporate that with the current focus.

**Manuscript changes:**

We added a sentence in Section 3.1 clarifying that: *"Integrating the DelayFixed adapter replaces $sm(t)$ with $sm(t-1)$ in the coupled equation, thus delaying the effect of changes by one timestep."*